# Delineating incised stream sediment sources within a San Francisco Bay tributary basin

Paul Bigelow[1], Lee Benda[2], and Sarah Pearce[3]

[1] Bigelow Watershed Geomorphology, Oakland, California, USA
[2] Terrain Works, Mount Shasta, California, USA
[3] San Francisco Estuary Institute, Richmond, California, USA

*Correspondence to*: Paul Bigelow (paul@bigwatershed.com)

**Abstract.** Erosion and sedimentation pose ubiquitous problems for land and watershed managers, requiring delineation of sediment sources and sinks across landscapes. However, the technical complexity of many spatially explicit erosion models precludes their use by practitioners. To address this critical gap, we demonstrate a contemporary use of applied geomorphometry through a straightforward GIS analysis of sediment sources in the San Francisco Bay Area in California, USA, designed to support erosion reduction strategies. Using 2 m LiDAR DEMs, we delineated the entire river network in the Arroyo Mocho watershed (573 km$^2$) at the scale of ~30 m segments and identified incised landforms using a combination of hillslope gradient and planform curvature. Chronic erosion to the channel network was estimated based on these topographic attributes and the size of vegetation, and calibrated to sediment gage data, providing a spatially explicit estimate of sediment yield from incised channels across the basin. Rates of erosion were summarized downstream through the channel network, revealing patterns of sediment supply at the reach scale. Erosion and sediment supply were also aggregated to subbasins, allowing comparative analyses at the scale of tributaries. The erosion patterns delineated using this approach provide land use planners with a robust framework to design erosion reduction strategies. More broadly, the study demonstrates a modern analysis of important geomorphic processes affected by land use that is easily applied by agencies to solve common problems in watersheds, improving the integration between science and environmental management.

## 1 Introduction and Objective

Channel incision is a common erosional response to natural or anthropogenic forcing that poses challenges to watershed management across the globe (Schumm et al. 1984, Schumm 1999, 2007). Incised channels (inner gorges, arroyos, gullies, ravines, etc.) are often created by headward incision of the channel network in response to local or regional base-level lowering (Begin et al. 1981, Schumm 1993), tectonic uplift (Burnett and Schumm 1983), or disturbances that change the relative balance between sediment transport and supply (Schumm et al. 1984, Schumm 1999). For example, incision caused by humans can include urbanization that greatly increases runoff and sediment transport (Booth 1991), or dams and gravel mining that reduce sediment supply (Kondolf 1997, Surian and Rinaldi 2003). Examples of natural causes of incision include change to a wetter climate that increases runoff (Balling and Wells 1990) and catastrophic events that increase sediment supply from volcanic eruptions (Simon 1992), extreme precipitation (Miller and Benda 2000), or following wildfire (Benda et al. 1998). Characteristically, channel incision continues until a new equilibrium grade is achieved, then the channel widens by eroding and failing oversteepened banks, and aggradation begins (Schumm et al. 1984). In some environments, incised channels are part of a natural alternating cycle of aggradation and degradation in response to episodic sediment supply (Bull 1997). Channel incision creates a variety of problems including destruction of valley bottoms (arable land) and increased sediment yield aggrading downstream reaches (Patton and Schumm 1975, Poesen et al. 2003).

Delineating the extent of channel incision across a watershed or landscape and quantifying erosion from such sources are necessary to design erosion and channel sedimentation abatement measures (Heede 1974, Schumm 1984). Mapping gullies using remote sensing began in the 1970s (e.g., Patton and Schumm 1975) but advancing digital technology in the 21st century now allows for ever more detailed mapping using geographic information systems (GIS) and digital elevation models (DEM). Mapping the extent of incised channels can involve visual detection (hand digitizing) in conjunction with delineating a zone around channels based on channel size using GIS (called buffering) (e.g., Perroy et al. 2010) or more automated approaches using digital terrain analysis (Evans and Lindsey 2010, Castillo et al. 2014) or object-oriented classification of gullies (Shruthi et al. 2011, Johansen et al. 2012). Quantifying erosion rates along incised channels often involves calculating the surface elevation difference between current and pre-gully DEMs (Perroy et al. 2010, Evans and Lindsay 2010) or using a time series of DEMs (Brasington et al. 2000, Fuller et al. 2003, Martınez-Casasnovas 2003, Wheaton et al. 2010, Picco et al. 2013). Spatially distributed estimates of sediment yield from incised channels often require highly parameterized and calibrated models (e.g., Van Rompaey et al. 2001, Pelletier 2012). The technical complexity of such approaches precludes their use by many watershed, land, and resource managers (Guertin and Goodrich 2011) and consequently such agencies often resort to qualitative evaluations or best professional judgment.

In this case study of a San Francisco Bay tributary, we delineate chronic annual erosion from the sides of incised channels and estimate sediment yield at multiple scales using an approach readily understandable and accessible to planners that improves the spatially explicit representation of sediment supply through the channel network. We use a topographic index that combines slope and planform curvature to predict generic erosion potential (GEP) through the process of shallow failures on planar and convergent slopes (Miller and Burnett 2007, Benda et al. 2011). GEP values are calibrated to sediment yield from gage data and then reported to the channel network and aggregated downstream. Using a terrain mapping platform (NetMap) (Benda et al. 2007, 2015a, Barquin et al. 2015), we derived a digital river network directly from a 2 m light detection and ranging (LiDAR) digital elevation model (DEM) and couple it to the terrestrial landscape via flow direction and accumulation grids. Each ~30 m reach is coupled to its local hillslope contributing area (although confined to the incised channel landform in our study application) and GEP and sediment yield values are transferred to the reaches by that contributing area (drainage wings).

The study objective was to delineate and quantify the spatial distribution of chronic sediment supply from incised channels across the Arroyo Mocho watershed in support of erosion reduction strategies. Ultimately, this study demonstrates a straightforward GIS analysis of important geomorphic processes (erosion and sedimentation) often impacted by land use that can be easily applied by watershed and land managers. Within the context of this special issue, our study provides a contemporary example of applied geomorphomety, one designed to increase communication between science and resource planning.

## 2    Basin Physical Characteristics and Background

Arroyo Mocho basin drains 573 km$^2$ of the Livermore Valley and is tributary to Arroyo de la Laguna, which joins Alameda Creek and drains to the San Francisco Bay, California (Fig. 1). Basin elevations range between 60 and 1200 m and mean annual precipitation averages 428 mm (Prism 2012). Land use in the watershed includes a mix of urban, residential, and commercial areas concentrated on the Livermore and tributary valley floors, with agriculture (e.g., vineyards) in the lower foothills and rural areas in the uplands. Annual grasses are the dominant vegetation in the watershed. Sparse patches of remaining riparian

vegetation in the basin include willows, cottonwoods, occasional oaks, alkali sink scrub, and herbaceous scrub (Stanford et al. 2013), while conifers can be found in the higher elevations. The Livermore Valley is a large tectonically formed depression (pull-apart basin) infilled with late Tertiary and Quaternary alluvial sediment (Graymer et al. 1996, Helley and Graymer 1997). Major tributaries to Arroyo Mocho include Alamo, Tassajara, Cayetano, Altamont, Arroyo Seco, and Las Positas Creeks (Fig. 1).

Historically, broad distributary fans formed where these tributaries entered the Livermore Valley floor and lagoon ponding occurred at the western distal end of the valley (Williams 1912), likely in response to massive landsliding from fault rupture (Ferriz 2001). The fans, valley floor, and marsh were channelized and drained in the late 1800s (Williams 1912) creating a direct conduit to Arroyo de la Laguna (Fig. 1). The region has experienced several cycles of Holocene and earlier incision and alluviation from climatic and tectonic forcing (Sloan 2006) and more recently from land use changes and channelization (Rogers

1987, Mero 2015, Williams 1912). Headward growth of stream valleys and canyon cutting (arroyos and gullies) were the dominant geomorphic agents in the basin (Hall 1958), where sediment from these incised channels is now chronically supplied by steep eroding banks (Bigelow et al. 2012a). At larger drainage areas, the incision created continuous arroyos cut into valley fill that are more permanent features on the landscape. At smaller drainage areas, the incision created discontinuous or patchy gullies cut into both narrow valley fills and colluvial hillslopes that are likely more ephemeral features. The engineered flood

control channels on the Livermore valley floor (Fig. 1) are currently aggrading in some areas, providing the primary motivation for this study to inform sediment reduction efforts by local government jurisdictions.

In addition to bank erosion from incised channels, mass wasting processes also occur in the uplands, primarily earthflows (Davenport 1985, Wentworth et al. 1997, Roberts et al. 1999, Majmundar 1991 and 1996). The southern hills are underlain by

the hard meta-sedimentary rocks of the Cretaceous Franciscan Formation with patchy outcrops of the Plio-Pleistocene Livermore Gravels. Here, the steep topography is dominated by deep-seated landslides or earthflows, most of which are old and no longer active. The eastern hills are underlain by the Cretaceous Great Valley Sequence and Miocene sedimentary units that tend to have steep slopes prone to earthflows. The northern hills are comparatively gentler, underlain by the Livermore Gravels and Miocene sedimentary rocks that produce clay rich soils prone to earthflows.

**3       Methods**

### 3.1     Definine Sediment Source Area

Channels in the Arroyo Mocho watershed are most often characterized by arroyo or gully forms: an incised topography within a broad valley floor, with steep and occasionally bare eroding banks. The raw banks appear to dominate the chronic annual supply of sediment to channels in the study basin, and thus represent the main source of sediment to the aggrading channels

downstream. In general, the low gradient valley floors above arroyo banks cannot topographically erode or deliver sediment to the channel. In the steeper upland channels bordered by colluvial hillslopes (i.e. no valley floor), most of the sediment production occurs on the channel banks and hillslope areas adjacent to the channel, including the toes of earthflows that intersect the channel; hillslopes farther from the channel do not appear to deliver sediment on an annual basis. Fig. 2 shows typical cross sections of valley floor and hillslope channels. Based on these observations, we confined the analysis of erosion sources to areas

adjacent to channels (e.g., the arroyo or gully landform). To use a buffer width that scales regionally with drainage area, we used a GIS buffer width of 6 times the total bankfull channel width that captures the steep eroding banks of the incised channel form and areas immediately adjacent to the channel (Fig. 3). To estimate bankfull channel width, we used a San Francisco Bay Area regional regression relationship based on drainage area (Dunne and Leopold 1978):

$$\text{bankfull channel width (m)} = 3.3494 \text{ (drainage area [km}^2\text{])}^{0.3737} \hspace{3cm} (1)$$

In comparison to field measurements downstream from Arroyo Mocho, Bigelow et al. (2008) found the regional regression slightly over estimated actual bankfull width. The channel width estimate used here is only to create a buffer capturing the area around incised channels, not to define channel edges. We checked the fit of several buffer sizes (2, 4, and 6 times total bankfull width) on the shaded relief DEM (Fig. 3), air photos, and in the field, and found the 6 times total bankfull width buffer was best suited to maximize the inclusion of both eroding banks of incised channels and hillslope areas that contribute sediment annually (e.g. toes of earthflows), but exclude areas that are not directly connected to channels. A small area of valley floor or hillslope of generally low erosion potential beyond the channel edge is included in this buffer. This buffering to capture the incised channel network still provides the most reasonable solution, considering the massive effort needed for alternatives to either digitize incised channels over a 573 km$^2$ watershed or develop code to automate the procedure. The defined erosion source area (98 km$^2$) is hereafter referred to as the buffered incised channel network.

### 3.2    Primary Steps of the Analysis

We delineated and quantified the spatial distribution of chronic sediment supply from banks along incised channels that can include earthflow toes (can include the processes of small shallow failures and raveling) across the Arroyo Mocho watershed using a terrain mapping platform (NetMap) that has been applied in similar applications elsewhere (Benda et al. 2007, 2009, 2011, 2015, Bidlack et al. 2014, Barquin et al. 2015, Flitcroft et al. 2015). The primary tasks in the analysis included:

1. Generate a digital and attributed stream layer within the virtual watershed.

2. Estimate erosion potential of incised channel banks using a topographic index that includes hillslope gradient and planform curvature (topographic convergence).

3. Modify erosion potential based on vegetation size.

4. Convert erosion potential to sediment yield using river gage data.

5. Aggregate erosion predictions at various scales: buffered channel reaches, subwatershed, tributary watershed, channel network.

6. Estimate sediment storage potential based on channel constraint and stream power.

### 3.3    DEM and Stream Network

An attributed digital stream network was derived from the DEM using a series of algorithms described by Miller (2002). The following major steps were followed in the construction of the DEM and extraction of the channel network:

1. *DEM*. The 2 m DEM was compiled and resampled from a 0.3 m DEM for the majority of the watershed within the dominant jurisdictional boundary (Alameda County) and a 3 m DEM for small portions of the watershed that lie in other

counties, all derived from LiDAR data collected by the U.S. Geological Survey. The objective here was to maintain information from the most precise and accurate data in creation of a single, contiguous DEM, while also avoiding creation of breaks in elevation or derivatives of elevation (gradient, curvature) at seams between the different data sources.

2.  *Topographic attributes*. Topographic attributes for network extraction were based on unaltered elevation data, prior to drainage enforcement and hydrologic conditioning. The attributes calculated are surface gradient, plan curvature, and the contour length crossed by flow out of a DEM cell, which is used to calculate local contributing area.

3.  *Channel Initiation*. Channel initiation points were defined using slope-dependent drainage area threshold criteria (e.g. Montgomery and Dietrich 1992, Dietrich et al. 1993) based on flow accumulation and surface slope, plan (contour) curvature, and flow length over which these criteria are met (Miller 2002, Miller et al. 2015).

4.  *Hydrologically conditioned DEM*. A hydrologically conditioned DEM was created where the flow paths out of all closed depressions are identified using a combination of depression filling (Jenson and Domingue, 1988) and carving (Soille et al., 2003).

5.  *Flow and channels*. Flow accumulation was calculated to identify all channel initiation points, and trace all channels. The D-infinity algorithm (Tarboton, 1997) was applied to calculate flow accumulation values down to channel initiation points. Channels were then traced downstream from these points using D-8 flow directions (Jenson and Domingue, 1988) to preclude dispersion of channelized flow. D-8 flow directions are chosen using a combination of steepest descent and largest plan curvature (Clarke et al., 2008). Using algorithms by Miller (2002), the channel network was divided into homogeneous reaches with end-point positions selected that minimize variance of channel gradient, valley width, and drainage area within a reach, while keeping mean reach lengths no more than 80 m (ave 30 m); each reach was attributed with various topographic measures, including channel slope, valley width, drainage area, etc. (Miller 2002).

6.  *Refining Network*. Channel traces were smoothed to provide better-placed channel centerlines and more accurate estimates of channel length and gradient. The delineated channel network was validated using a combination of local knowledge, field observations, and high-resolution optical imagery.

More details on methods to derive the channel network from the DEM and algorithms used can be found in Miller et al. (2015), Clarke et al. (2008), and Miller (2002).

## 3.4     Estimating Erosion Potential from Incised Channels and Other Sources

To provide a relative estimate of erosion potential within the buffered incised channel network of the Arroyo Mocho watershed, we used a topographic index called GEP (Miller and Burnett 2007, Benda et al. 2011) that combines slope steepness with slope convergence, recognized topographic indicators of shallow landsliding, gully erosion, and sheetwash (Dietrich and Dunne 1978, Sidle 1987, Montgomery and Dietrich 1994, Miller and Burnett 2007, Parker et al. 2010). Slope steepness is a fundamental control on erosion potential, while convergence causes surface and subsurface flow to become concentrated and contribute to

erosion potential. GEP (Eq. 2) is calculated from topographic attributes of slope gradient and topographic convergence (planform curvature) derived from the DEM:

$$GEP = S \cdot a_L/b \qquad\qquad (2)$$

where GEP is the generic erosion potential, S is slope gradient (m/m), $a_L$ is the local contributing area to a DEM pixel, and b provides an estimate of the total contour length crossed by flow exiting a pixel. Local contributing area $a_L$ is calculated using the D-infinity flow direction algorithm (Tarboton 1997), which allows downslope dispersion. The flow direction for each pixel is calculated using each of eight triangular facets defined by a DEM grid point and the eight adjacent points, where values range from 0 - 8. For each facet having flow out of the pixel, we use the projection of flow direction on the exterior facet edge as a measure of contour length crossed by flow exiting the pixel from that facet. The projection lengths are summed over all edges with outgoing flow to provide an estimate of contour length b for flow exiting the pixel, where values range from 0 - 4. Contour length for planar flow is one pixel width, for divergent flow it is greater than one pixel width, and for convergent flow it is less than one pixel width. The ratio $a_L/b$ therefore incorporates effects of topographic convergence (Miller and Burnett 2007). Multiplying by slope provides a basin-wide measure of erosion potential, similar to a slope area product, but using only the local contributing area. Thus GEP is a dimensionless relative index of erosion potential with most values within a watershed ranging from 0 – 1, where larger values correspond to steeper, more convergent topography prone to higher landslide densities, surface erosion, and higher gully-initiation-point densities (Miller and Burnett 2007).

Local contributing area is used for erosion potential rather than the entire upslope drainage area because (1) conceptually, pore pressures measured during storms in shallow soils correlate poorly with topography (e.g., Dhakal and Sullivan 2014), but convergent areas tend to exhibit persistently high water content, deeper soils, and are highly responsive to rainfall events, and (2) empirically, local contributing area has been shown to better predict shallow failures than total contributing area in several studies (Miller 2004, Miller and Burnett 2007). While upslope drainage area is often used as a surrogate for subsurface flow, most locations on the hillslope receive contributions from a small proportion of the upslope contributing area due to the low velocity of subsurface flow (Barling et al., 1994, Beven and Freer 2001, Borga et al. 2002).

We specifically use GEP as a relative index of erosion potential for shallow failures along steep banks of incised channels in the Arroyo Mocho study area. GEP is not used to estimate erosion potential for downcutting the channel bed or bank erosion of outside river bends. As mentioned earlier, the incised channels in the Arroyo Mocho watershed do not appear to be actively downcutting, rather sediment is now supplied to most channels by shallow slides and failures of oversteepened banks (channel widening), a common stage in the evolution of incised channels following downcutting (Schumm et al. 1984).

Because the analysis was confined to the buffered incised channel network, we assume that all sediment eroded from the inner gorges, arroyos, and gullies is delivered directly to the channel network, as confirmed during two days of field observations across the watershed. When practitioners want to include other forms of erosion further from the channel, the proportion of sediment delivered to the channel should be estimated based on different topographic attributes (e.g., Mitasova et al. 1996, Miller and Burnett 2007, Cavalli et al. 2013), which can be automated within the NetMap terrain mapping platform (Benda et al. 2007) and other similar approaches.

### 3.5    Modifying Erosion Potential by Vegetation

We observed that arroyo and gully bank erosion was often reduced by vegetation in the Arroyo Mocho basin, where larger and denser vegetation created stable channel bottoms and banks. Reaches that had little to no vegetation had more exposed, actively eroding banks compared to reaches that had shrubs or trees established on the banks. Where present, the effect of shrubs and trees in reducing erosion in these arroyo channel systems is particularly pronounced because there is little organic groundcover other than the dominant vegetation of annual grasses. Vegetation reduces erosion by lessening raindrop impact, providing increased soil strength through the root network, and thus reducing surface erosion, rill erosion, and shallow bank slumps and slips (e.g., Thornes 1985, Thorne 1990, Prosser and Dietrich 1995, Simon and Darby 1999, Abernethy and Rutherford 2001, Micheli and Kirchner 2002). In addition, increasing tree age (and thus rooting extent and depth) is related to increasing stability of the soil (Sidle 1987), and tree height and canopy width are proportionally related to rooting width and depth (e.g. McMinn 1963, Smith 1964, Tubbs 1977, Gilman 1988). These relationships provide a basis for using tree height as a proxy for root spread and thus soil stability as described below.

While approaches linking vegetation and associated rooting strength to specific types of erosion have been developed for highly localized scales (e.g., Roering et al. 2003), such empirically-based quantitative approaches to reduce erosion potential using remotely sensed vegetation attributes have yet to be developed, consequently, many erosion models simply use broad categories of land cover. For example, the Universal Soil Loss Equation (USLE) (Wischmeier and Smith 1978), the revised USLE (RUSLE) (Renard et al. 1997), and similar approaches (e.g., Booth et al. 2014) simply use a generic reduction (C-factor) based on classes of vegetation cover (e.g., forest, scrub, etc.) to adjust erosion predictions over vast areas regardless of the individual size of vegetation within the categories. Other approaches infer similar generalized relationships, for example, Pelletier (2012) assumed a linear relationship between vegetation (leaf cover) and sediment detachment. We also assume the occurrence of vegetation reduces erosion potential, but use a relationship that includes both the effects of vegetation cover and rooting width and depth in each 2 m grid cell, using tree height as a proxy for root spread and related soil stability. During one day of field observations across the watershed, we observed less erosion on banks with taller trees (and larger root spread) compared to banks with smaller trees and shrubs. Here, we visually estimated the average riparian vegetation height on a given bank and the proportion of the bank eroded, the inverse of the latter gives an estimate of erosion reduction. These observations were plotted to derive the best fit equation (Eq. 3, Fig. 4) that governs how erosion potential is reduced by vegetation height:

Erosion reduction = 0.1906 ln(vegetation height in m) + 0.136                    (3)

To reduce GEP based on this relationship, a vegetation height grid (2 m) was created in GIS using the first (representing the tallest vegetation) and last (representing the ground surface) return LiDAR points. An erosion reduction grid was then created by multiplying the vegetation height grid by Eq. 3. Here, a tree canopy height of 1, 21, or 42 m (maximum tree height) would produce erosion reduction values 0.14, 0.72, and 0.85, respectively (i.e. 14, 72, and 85% erosion reduction). The resulting erosion reduction grid was subtracted from the initial GEP grid to estimate erosion sources. For example, where a grid cell has a GEP value of 1 and a corresponding erosion reduction value of 0.5, the resulting modified GEP would be 0.5. Where there was no vegetation, GEP values were unchanged. Maximum GEP reduction due to the greatest vegetation height was 0.85 (85%). We were unable to obtain raw LiDAR for Santa Clara County, so it was not possible to create a vegetation reduction grid for this small southeastern portion of the watershed at the headwaters of Arroyo Mocho canyon (Fig. 1). In the absence of more quantitative techniques, this method provides an incremental improvement on previous approaches. Where as an entire class of

vegetation covering square kilometers would get assigned a single erosion reduction factor (C factor) using USLE and RUSLE and similar approaches, here each 4 m$^2$ pixel gets assigned a reduction factor based on vegetation height as a proxy for root spread and associated soil or bank stability.

### 3.6    Conversion to Sediment Yield

To provide a more meaningful view of spatially explicit erosion across the watershed, we converted the dimensionless GEP index to sediment yield following approaches used by GMA (2007) and Benda et al. (2011). To limit assumptions, the independently estimated sediment yield rate is linearly scaled with GEP values. High values of GEP represent higher erosion rates and lower values of GEP represent lower erosion rates. GEP is converted to sediment yield by multiplying each GEP grid cell by the following conversion factor (Eq. 4):

$$\text{conversion factor} = \frac{\text{mean sediment yield rate at Verona Gage}}{\text{mean GEP buffered channel network}} \qquad (4)$$

where mean GEP along the buffered channel network is the mean of all GEP grid cells within the buffered channel network (e.g., GMA 2007, Benda et al. 2011). A previous study identified an average sediment yield of 155 tonnes km$^{-2}$ yr$^{-1}$ for the entire drainage area (573 km$^2$) using sediment data collected from 1994 to 2006 (suspended and bedload) at the Verona Gage on Arroyo de la Laguna (Bigelow et al. 2008)(Fig. 1). In the current application that restricts the analysis to the buffered incised channel network (drainage area of 98 km$^2$), an average sediment yield rate of 906 tonnes km$^{-2}$ yr$^{-1}$ was calculated using the same sediment gage data. The conversion to sediment yield does not account for any deposition within the channels and therefore underestimates the spatially explicit sediment supply rates, but likely corresponds to the correct order of magnitude within the context of sediment budgeting technology (e.g., Reid and Dunne 1996).

### 3.7    Spatial Distribution of Erosion at Various Scales and through the Channel Network

To evaluate the spatial distribution of chronic sediment sources from incised channels across the study basin, we calculated, aggregated, and mapped GEP and sediment yield at four scales: pixel, buffered reach area, subwatershed, and tributary basin. We also estimated the specific sediment yield for each reach to illustrate how sediment yield varies downstream through the channel network. The total sediment yield value at the watershed outlet equals the basin average sediment yield. To estimate sediment supplied to each reach (specific sediment yield), the buffered channel network was discretized to define the buffered drainage area on each side of the channel reach, or drainage wings. Using algorithms within NetMap, the total GEP and sediment yield within the drainage wings were attributed to each segment (reach) of the channel network. To estimate the specific sediment yield at a given stream segment, GEP and sediment yield was cumulatively added moving downstream and divided by total upstream drainage area of the buffered incised channel network. The delineation of erosion from incised channels was checked with direct field observations and by draping erosion predictions over air photos and checking predictions in the field.

### 3.8    Sediment Storage Potential

As indicated previously, sediment supply from the incised Arroyo Mocho channel network is aggrading portions of the engineered flood control channels on the Livermore Valley floor.  Sediment supply and aggradation of the flood control channels could be reduced by promoting sediment storage at upstream locations, for example by reconnecting channels to floodplains. To help land managers identify ideal upstream locations for sediment storage, we developed a relative sediment storage potential index (Eq. 5) that is calculated for each stream reach:

$$\text{storage potential index} = \frac{\text{valley width index}}{\text{stream power index}} \tag{5}$$

where the stream power index is drainage area * gradient, the valley width index (e.g. Grant and Swanson 1995) is valley width (at 2 x bankfull depth) divided by channel width. We multiply the index by a scale factor of 0.001 to keep the values between 0 – 100. The drainage area used here for stream power includes the entire upstream watershed area draining to a reach, not the drainage area of the buffered incised channel network. Channel width (Eq. 1) and bankfull depth (Eq. 6) were estimated using regional regressions (Dunne and Leopold 1978):

$$\text{bankfull depth (m)} = 0.3593 \, (\text{drainage area } [\text{km}^2])^{0.3593} \tag{6}$$

In comparison to field measurements downstream from Arroyo Mocho, Bigelow et al. (2008) found the regional regression estimates were similar and occasionally slightly lower than actual bankfull depth. However, the bankfull depth estimate is only used to provide some height above the channel that scales regionally with drainage area at which to measure valley width. For all reaches in the stream network, the stream gradient, drainage area, and valley width were extracted from the DEM using algorithms within NetMap (Miller et al. 2002, Benda et al. 2007) and attributed to the stream network layer. Stream power reflects the ability of a channel to transport or store sediment, where streams with higher stream power have less opportunity to create large in-channel storage reservoirs in contrast with streams of lower power that can store sediment. The valley width index reflects the potential width of the floodplain for sediment storage. For example, applying the index (Eq. 3) to a channel reach draining an area of 22 km$^2$, with a channel gradient of 0.007 m/m, valley width of 424 m, channel width of 11 m, and the scale factor 0.001 would produce a relative storage potential index of 0.24. Index values ranged from 0 – 90, with most values less than 1. We calculated the sediment storage potential index with valley widths at 1- 3 times the bankfull depth, and found valley width at 2 times the bankfull depth produced the most varied and best results in our basin. In addition, sediment storage potential indices using valley widths at greater than 2 times the bankfull depth could risk identifying channels with high storage potential, but so deeply incised they cannot be reconnected to floodplains.

## 4        Results and Discussion

### 4.1        Field and Air Photo Observations of Erosion

Viewing the GEP layer draped over high resolution aerial imagery along roughly 50% of the channel network, we consistently observed steep eroding banks (bare of vegetation) in areas with high GEP values throughout the watershed. Similar observations were confirmed during 2 days of fieldwork, where additionally we consistently observed much more stable banks with vegetation in areas with lower GEP values (Table 1, Fig. 5, also see extensive photo documentation in Bigelow et al. 2012a). In the field, higher GEP values generally corresponded to steeper more convergent terrain, while lower GEP values corresponded to flatter and divergent terrain. These observations qualitatively indicate that GEP provides reasonable estimates of relative erosion within a watershed. To quantitatively access the accuracy of GEP requires multiple long-term sediment gages within a watershed, which is not yet possible in this watershed. However, when comparing erosion predictions to field inventories in the Oregon Coast Range, the index performed better than hillslope gradient alone or other available erosion models (Miller and Burnett 2007). As mentioned previously, GEP is a relative measure of erosion within a basin, which alone is highly useful to

**4.2      What is the spatial distribution of chronic erosion from incised channels at various scales across the watershed?**

The spatial distribution of erosion (GEP) and sediment yield reveals strong patterns at the scales of pixel, reach, subwatershed, and tributary watershed scales. Starting at the smallest scales, the spatial distribution of GEP and estimated sediment yield can be evaluated at the level of individual pixels (4 m$^2$) to identify discrete eroding banks (Fig. 5). To place the example shown on Fig. 5 in practical terms for land managers, the eroding bends shown are chronically contributing an estimated 10 tonnes of bank material per year, a little less than one dump truck worth of sediment. This estimate reflects a temporally averaged yield, however, erosion in the region is highly episodic (e.g., Ellen and Wieczorek 1982, Bigelow et al. 2008). Consequently, the actual temporal dynamics are that such a bank may retreat several meters in an extreme event yielding hundreds of tonnes to the channel, followed by many years of little or no erosion. Moving up to the buffered stream reach scale (mean 586 m$^2$), the spatial distribution of erosion can be used to highlight eroding bends and banks along entire valley segments (Fig. 5).

Scaling up to the subwatershed distribution of erosion (mean 2.7 km$^2$, Fig. 6) shows the most erosive valley segments are not isolated to a single larger basin, but are generally grouped into several steep or heavily incised areas across the entire Arroyo Mocho watershed. Such areas include steeper uplands or canyon areas where the channel impinges on and erodes high terraces, or where the toes of earthflows impinge on the channel. In addition to these steep erosive subwatersheds, other subwatersheds in the lower Tassajara and Cayetano basins also display higher estimated sediment yields (Fig. 6). These areas are prone to earthflows from clay rich expansive soils produced from the underlying Plio/Pleistocene and Miocene sedimentary units. At the subwatershed scale, the more erosive areas have estimated sediment yields roughly 3 – 8 fold higher than more stable areas (Fig. 6).

The spatial distribution of GEP and estimated sediment yield at the tributary basin scale (mean 51 km$^2$) varies considerably (Fig. 7). At this largest scale, the more erosive areas are concentrated in the steeper dissected basins of the southeastern and eastern watershed, primarily the upper Arroyo Mocho basins, Arroyo Seco, and Altamont Creek, where GEP values and estimated sediment yield are 3 fold higher than western areas at the basin scale (Fig. 7).

**4.3      What is the estimated specific sediment yield down through the channel network?**

The specific sediment yield for each reach is the buffered upstream sediment supply divided by the upstream buffered drainage area. Estimated specific sediment yields aggregated downstream illustrate how sediment yield varies through the channel network (Fig. 8). Similar to the spatially explicit distribution of GEP and sediment yield across the terrain (Figs. 5 - 7), this channel segment scale analysis illustrates higher sediment supply from the more dissected steep terrain of Arroyo Mocho canyon. We also aggregated the specific sediment yield downstream by reach and divided it by the total sediment yield at the basin outlet to show the percentage of the total sediment yield incrementally downstream through the channel network (Fig. 9).

**4.4      How Can Spatially Explicit Erosion Estimates Inform Sediment Reduction Efforts?**

**Prioritize sediment source control**. The spatial distribution of GEP and estimated sediment yield at the various scales across the terrain and through the channel network provides a physical basis for evaluating and prioritizing sediment reduction strategies within a large watershed or region. The spatial distribution of erosion at the subwatershed scale (Fig. 6) is perhaps the

most useful for prioritizing potential source control activities, where as the spatial distribution of sediment yield through the channel network (Fig. 7) can focus source control at a finer scale, showing which channels to focus on, rather than entire subwatersheds. After a subwatershed has been prioritized for source control, the reach and pixel scale maps of erosion (Fig. 5) can be used to target specific valley segments, reaches, and banks within the subwatershed. This spatially explicit representation

of erosion allows watershed managers to target limited funds to areas where they will achieve the most reduction in sediment supply. It should be noted that source control efforts would help reduce chronic sediment supply, but will not prevent massive aggradation from century-scale extreme events (e.g., Dettinger and Ingram 2013) that have previously filled valley bottom channels in the Alameda Creek watershed (Bigelow et al. 2008).

**Prioritize areas for sediment storage.** In combination with the spatial distribution of erosion (Figs. 5 – 9), watershed managers can also use other parameters extracted from the DEM to prioritize areas for sediment storage downstream of the most erosive areas. In our study basin, the Livermore Valley was historically highly depositional, where tributaries deposited sediment as broad coalescing fans across the valley floor (Williams 1912). Today, most valley floor channels are physically disconnected from their floodplain by channelization (engineered channels or ditches) and channel incision that migrated upstream from that

channelization (base level lowering) and other causes. Where there is sufficient space to allow channels to reoccupy floodplains, we estimated ideal locations for promoting sediment storage (e.g., reconnection of channels to floodplains). Using Eq. 5, sediment storage potential is estimated to vary considerably across the stream network, where certain portions of the mainstem streams are estimated to have a higher potential for sediment storage compared to other segments (Fig. 10). These estimations do not capture local forcing of sediment storage from channel constrictions (e.g. bridges) or tributary confluences that may exert

primary controls on aggradation. While valley floor channels with high potential for sediment storage are found throughout the watershed (Fig. 10), there are more rural undeveloped valley floors in the eastern and southern portions of the basin, and locations here may be more conducive to restoration projects. As an example application of the sediment storage potential index in this region, Fig 11. shows an ideal area for sediment storage in lower Cayetano Creek, where the channel could be plugged and diverted onto a new channel created near grade with the former floodplain, or with a series of ponds and plugs, where the

channel is widened (creating ponds) and the excavated material from the channel widening is used to plug the channel (e.g. Rosgen 1997). The sediment storage potential here is moderately high because the valley is wide and the channel gradient is low. In addition to sediment supply and sediment storage potential, practitioners should also consider the causes, current evolutionary stage, and history of channel incision within the watershed when determining the location of restoration projects to promote sediment storage (Schumm 1999). The Arroyo Mocho basin has undergone several cycles of incision from tectonic uplift (Sloan

2006), channelization of valley bottoms (Williams 1912), and changes in vegetation and runoff (Rogers 1988). Still, the incision cycles, history, and causes remain poorly understood and such knowledge would better target restoration locations and perhaps prevent or mitigate future cycles of incision from anthropogenic causes.

## 4.5     Adjustments and Considerations

Like most geospatial tools and models, this approach can be adapted and improved for specific applications and objectives. For

instance, erosion rates could be adjusted based on other factors not accounted for in this study such as higher erosion rates for weaker lithologies or precipitation gradients across large basins. As an example, we describe two potential improvements to refine estimates in our study basin to characterize longer term, decadal sediment yield and the bedload component. We also include other considerations for practitioners in their watersheds, including combining analyses of sediment supply with other concerns in a watershed.

**Decadal Episodic Sediment Supply.** The estimated sediment yield in this study represents an average condition using only topographic and vegetation attributes that characterize chronic (annual or persistent) sediment supply from bare steep channels banks. However, sediment supply in many landscapes is highly variable in both space and time resulting from episodic processes driven by interactions among storms, vegetation, and topography (Benda and Dunne 1997). In the San Francisco Bay region, episodic mass wasting triggered during El Niño storms can dominate decadal sediment supply (Ellen and Wieczorek 1982, Bigelow et al. 2008). For example, Arroyo Mocho is tributary to Alameda Creek, where a single flood event comprised 48% of the total load over a 13-year period (Brown and Jackson 1973). Episodic mass wasting sources in decadal sediment supply can be estimated through a more detailed sediment budget approach (e.g., Reid and Dunne 1996), or more simply by increasing the sediment yield in active mass wasting areas based on regional literature values. For example, two similar studies (GMA 2007, Bigelow et al. 2012b) used digitized maps of active earthflows and local literature values of earthflow rates to appropriately increase the sediment yield estimates at these discrete locations within the watersheds. This approach in part accounts for differences in lithologic erosion rates across the basin, as higher sediment yields from earthflows often occur in specific formations that produce clay rich soils (e.g. Keefer and Johnson 1983).

**Bedload Yield**. This study approach characterizes total sediment supply to the stream network without regard to the proportion of bedload to the overall yield. Some characterization of the chronic bedload yield variation throughout the stream network would better constrain source control efforts to areas that have a higher bedload yield aggrading valley floor flood control channels. Primary controls on bedload yield are typically drainage area and lithology, where the proportion of bedload generally decreases downstream due to particle attrition (abrasion and breakage) (e.g. Madej 1995, Benda and Dunne 1998) and attrition rates will also vary based on lithology, where harder rocks in a catchment produce a larger bedload component (Madej 1995, Turowski et al. 2012, Mueller and Pitlick 2013, O'Connor et al. 2014). While approaches characterizing bedload yield variation across a watershed can range in complexity (e.g., Dietrich and Dunne 1978, Collins and Dunne 1989, Benda and Dunne 1997, Madej 1995, O'Connor et al. 2014), a simplified approach is likely appropriate considering that limitations of sediment budgeting technology typically constrain estimates to the correct order of magnitude (e.g., Reid and Dunne 1996).

**Other Considerations**. This case study focuses on sediment supply primarily from incised channels, however, the approach can also be applied to most other forms of erosion including landslides, debris flows, and surface erosion across a watershed and including an estimate of sediment delivery (e.g., Miller and Burnett 2007, Benda et al. 2011). While we have focused on characterizing sediment supply and floodplains to identify potential restoration locations, many land managers have additional considerations when prioritizing stream restoration projects, including aquatic habitat, riparian habitat, and stream temperature. Such considerations can be incorporated using the same type of terrain analysis of additional topographic and vegetation attributes and other appropriate models and data. For example, intrinsic fish habitat potential can be estimated using channel slope, valley constraint (valley width/channel width), and mean annual flow (e.g., Burnett et al. 2007); large wood supply to streams can be estimated using a tree height layer and tree mortality rates (e.g., Benda et al. 2015, Flitcroft et al. 2015); and thermal loading to streams and effects of riparian shade can be estimated from a vegetation basal area and height layer and bare earth radiation calculations (e.g., Groom et al. 2011). Ultimately the various characterizations of intrinsic fish habitat, sediment supply, large wood supply, stream temperature, and other parameters of interest can be overlain and combined into a robust analysis to delineate and prioritize the most beneficial restoration locations across watersheds and landscapes (e.g., Benda et al. 2007).

# 5    Summary and Concluding Remarks

This case study shows how practitioners can delineate sediment sources from incised channels and prioritize potential sediment source control locations using the following primary steps:

1. Define the incised stream network.

2. Characterize erosion potential using a topographic index of slope and convergence as well as the presence and size of vegetation.

3. Convert erosion potential to sediment yield based on available gage data.

4. Scale up erosion potential and sediment yield from the pixel scale to reach, subtributary, and tributary, and major basin scales to observe variation in erosion potential and sediment supply at various scales across the watershed.

5. Estimate the specific sediment yield for each reach to illustrate how sediment yield varies downstream through the channel network.

6. Characterize sediment storage potential using a valley width index and stream power to identify potential areas to reconnect channels to floodplains.

The spatially explicit erosion modeling approach used here demonstrates a straightforward GIS analysis incorporating contemporary geomorphmetric techniques that only requires a DEM and vegetation height layer and characterizes erosion from incised stream banks and the toes of earthflows. The erosion model can also be used to characterize most other forms of erosion and sources further from the channel that require an estimate of sediment delivery (e.g., Miller and Burnett 2007, Benda et al. 2011). Accordingly, this approach should appeal to those seeking a simpler easily applied erosion model with wider applications compared to more complex models that are highly parameterized or limited to specific erosion processes. The approach can be adjusted for other factors influencing sediment supply (e.g. precipitation gradients, variation in lithology, episodic supply) and combined with analyses of additional topographic attributes and models to address additional concerns in a watershed such as fish habitat, riparian conditions, and stream temperature. Overall, this study illustrates a contemporary analysis of geomorphic processes that can be used by practitioners to solve common problems in watersheds.

**Acknowledgements.** The Zone 7 Water Agency funded an initial study that provided the basis for this paper. Dan Miller wrote and compiled the code used for DEM analysis used in NetMap for this study. Kevin Andras developed the various NetMap GIS layers. Jamie Kass and Julie Beagle helped obtain and process the LiDAR data and develop the vegetation height grid. Lester McKee provided a review of the initial Zone 7 study report. Lorenzo Marchi, two anonymous referees, and the associate editor Giulia Sofia provided thoughtful and constructive comments that improved the manuscript. We thank all of these people for their support of this project. We also express our sincere gratitude to David Bowie, for his beautiful and inspiring life and music that make "the stars look very different today".

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

**Table 1.** Summary of agreement between erosion (GEP) predictions and erosion observed on satellite imagery and in the field.

| Tributary Basin | GEP Value | Observed |
| --- | --- | --- |
| Lower Alamo Creek | High | High |
| Lower Alamo Creek | Moderate | Moderate |
| Lower Alamo Creek | Low | Low |
| Upper Alamo Creek | High | High |
| Upper Alamo Creek | Moderate | Moderate |
| Upper Alamo Creek | Low | Low |
| Lower Tassajara Creek | High | High |
| Lower Tassajara Creek | Moderate | Moderate |
| Lower Tassajara Creek | Low | Low |
| Upper Tassajara Creek | High | High |
| Upper Tassajara Creek | Moderate | Moderate |
| Upper Tassajara Creek | Low | Low |
| Cayetano Creek | High | High |
| Cayetano Creek | Moderate | Moderate |
| Cayetano Creek | Low | Low |
| Altamont Creek | High | High |
| Altamont Creek | Moderate | Moderate |
| Altamont Creek | Low | Low |
| Arroyo Seco | High | High |
| Arroyo Seco | Moderate | Moderate |
| Arroyo Seco | Low | Low |
| Lower Arroyo Mocho | High | High |
| Lower Arroyo Mocho | Moderate | Moderate |
| Lower Arroyo Mocho | Low | Low |
| Middle Arroyo Mocho | High | High |
| Middle Arroyo Mocho | Moderate | Moderate |
| Middle Arroyo Mocho | Low | Low |
| Upper Arroyo Mocho | High | High |
| Upper Arroyo Mocho | Moderate | Moderate |
| Upper Arroyo Mocho | Low | Low |
| Top Arroyo Mocho | High | High |
| Top Arroyo Mocho | Moderate | Moderate |
| Top Arroyo Mocho | Low | Low |

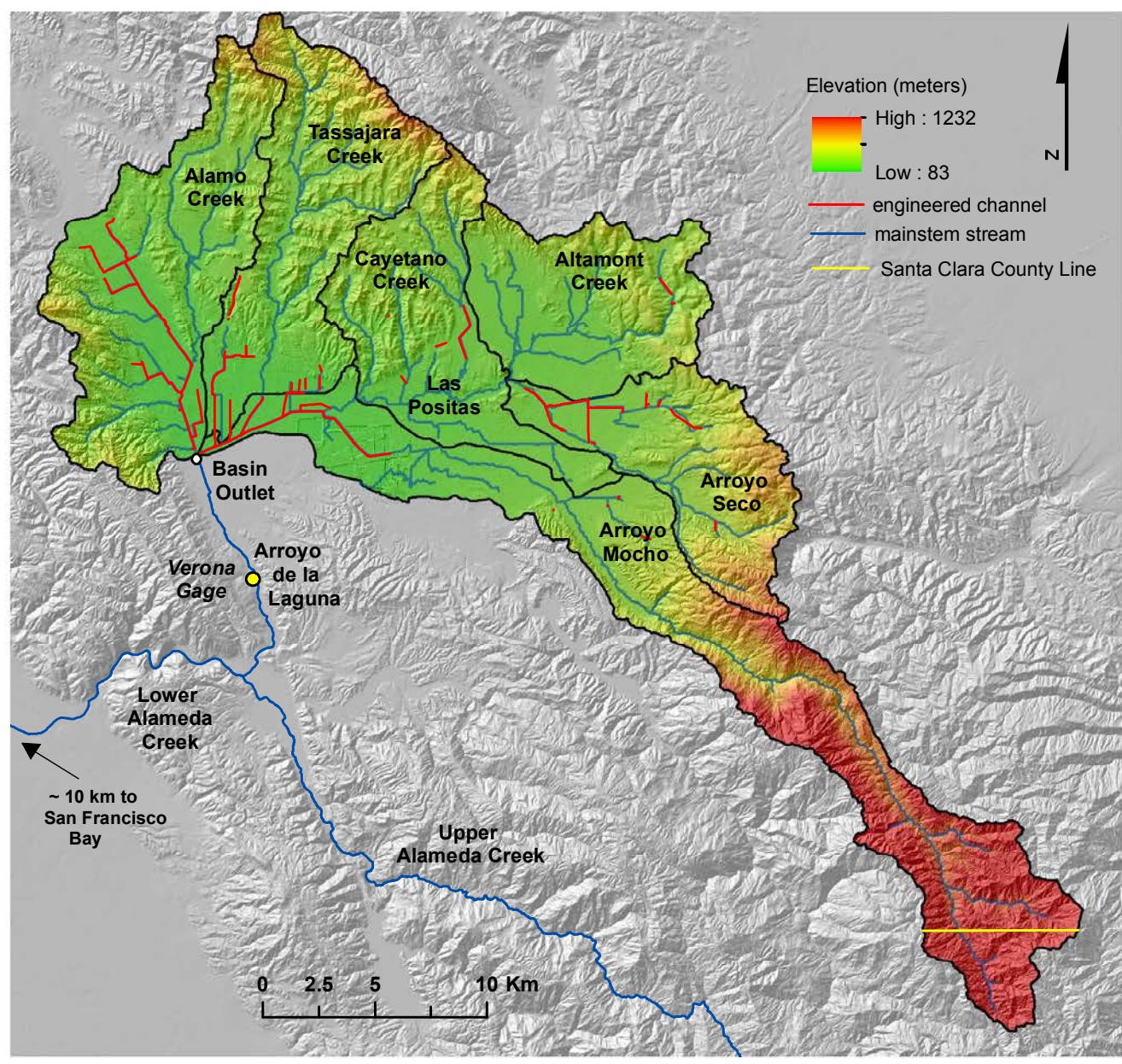

**Figure 1.** Arroyo Mocho watershed showing the six major basins, mainstem channels, and flood control channels.

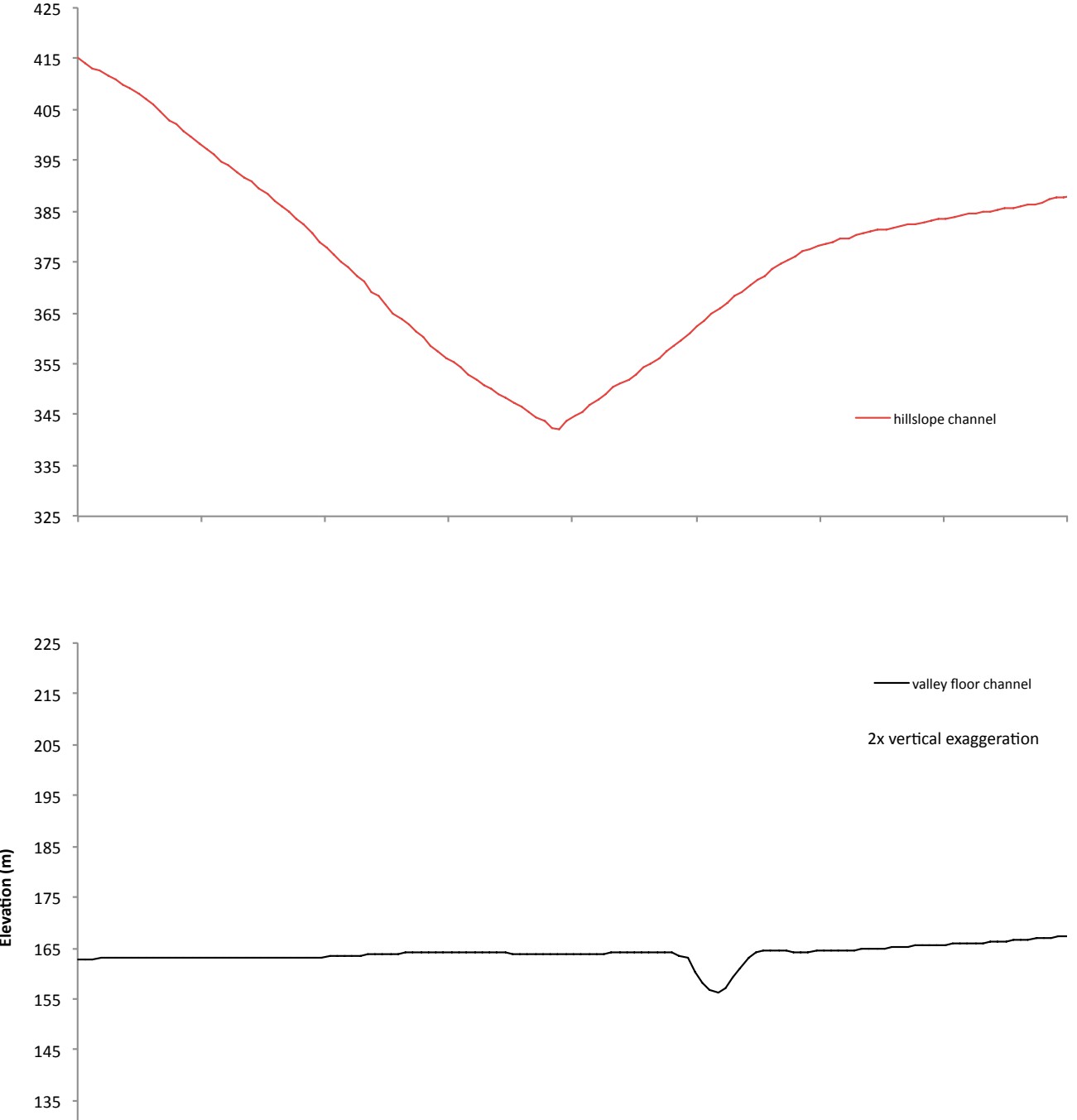

**Figure 2.** Typical topographic cross sections of a valley floor channel and hillslope channel from the Tassajara Creek tributary.

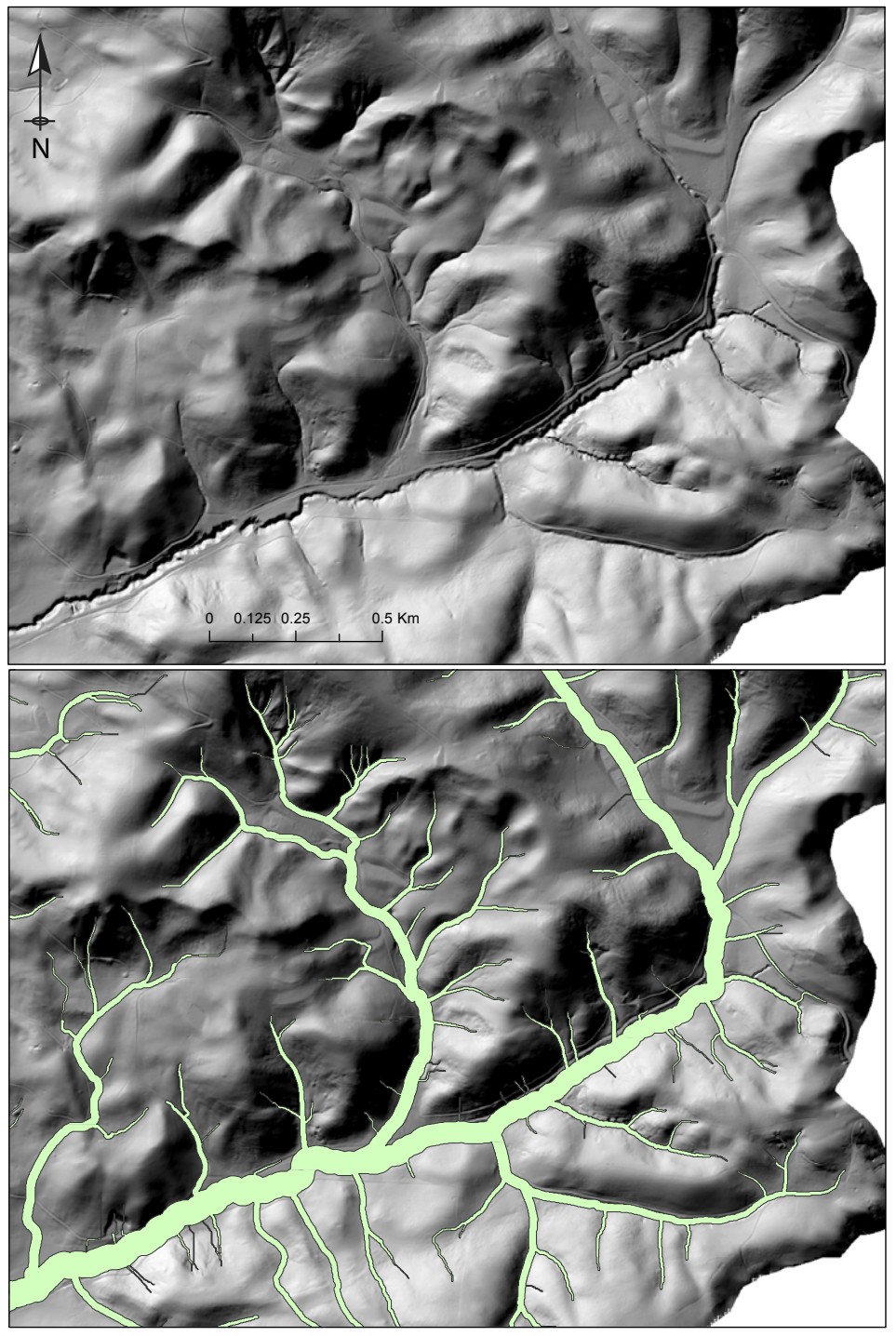

**Figure 3.** Example of the buffer used to define the sediment source area from incised channels in the upper Arroyo Seco basin.

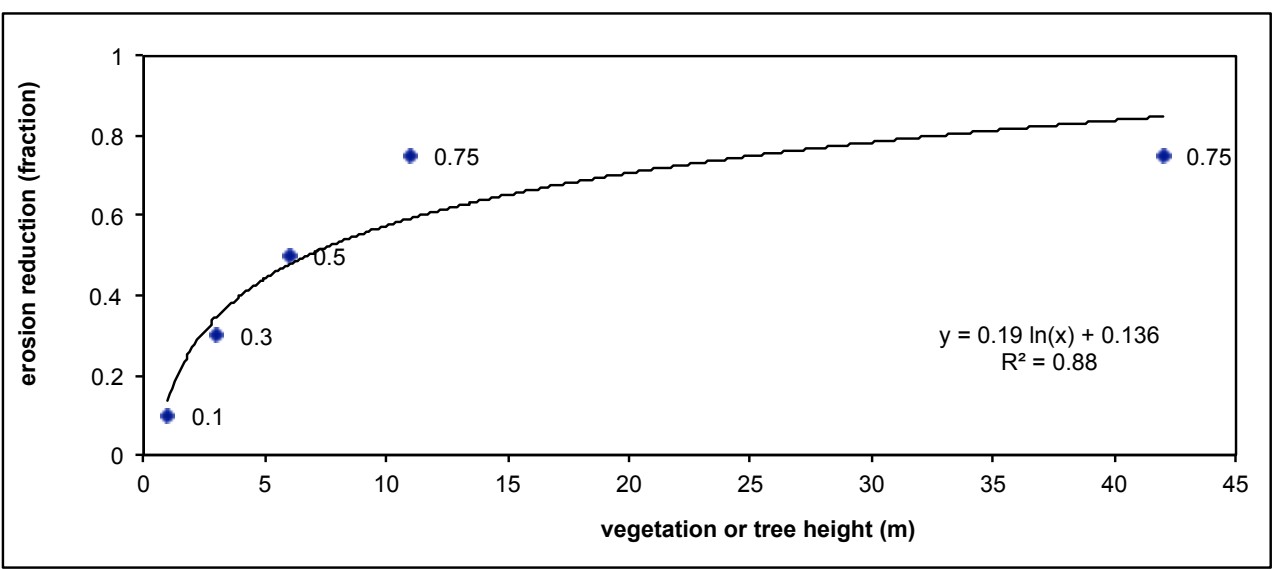

**Figure 4.** Relationship between tree height (as a proxy for root spread) and erosion reduction (bank stability) used to reduce GEP estimates. Relationship based on field observations.

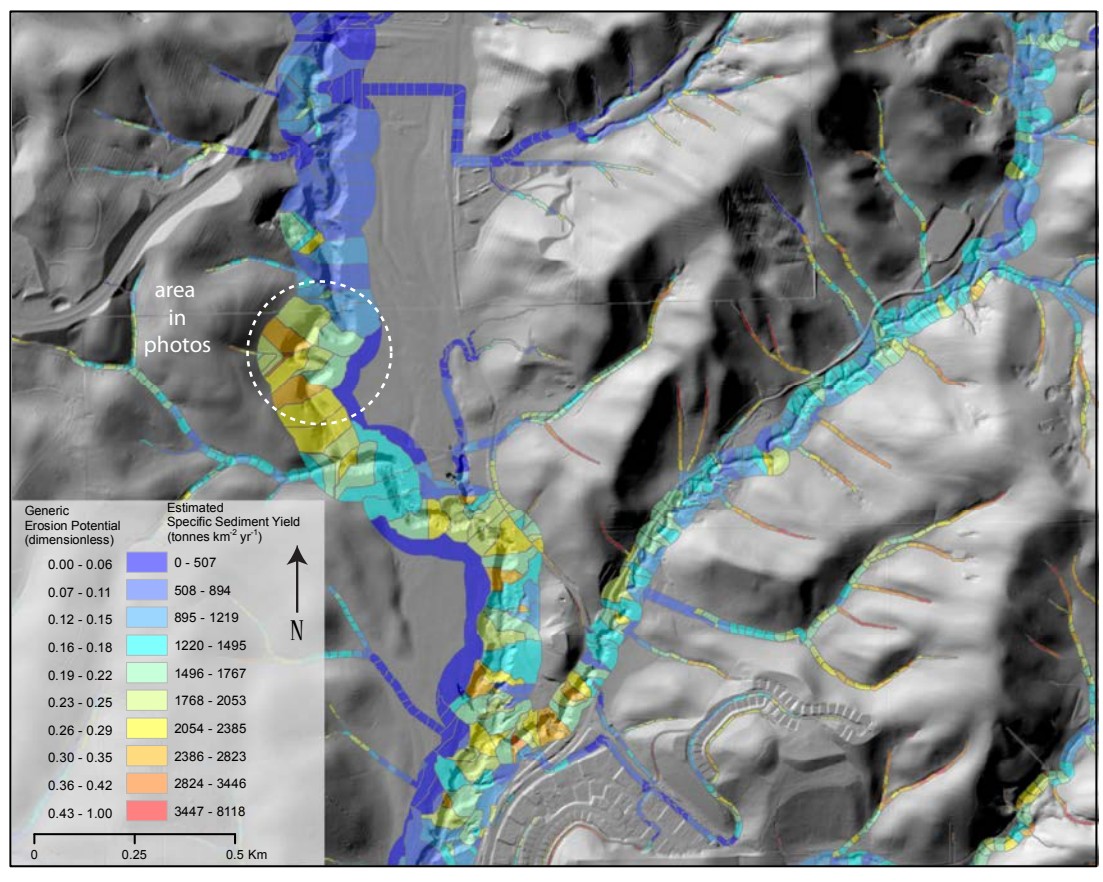

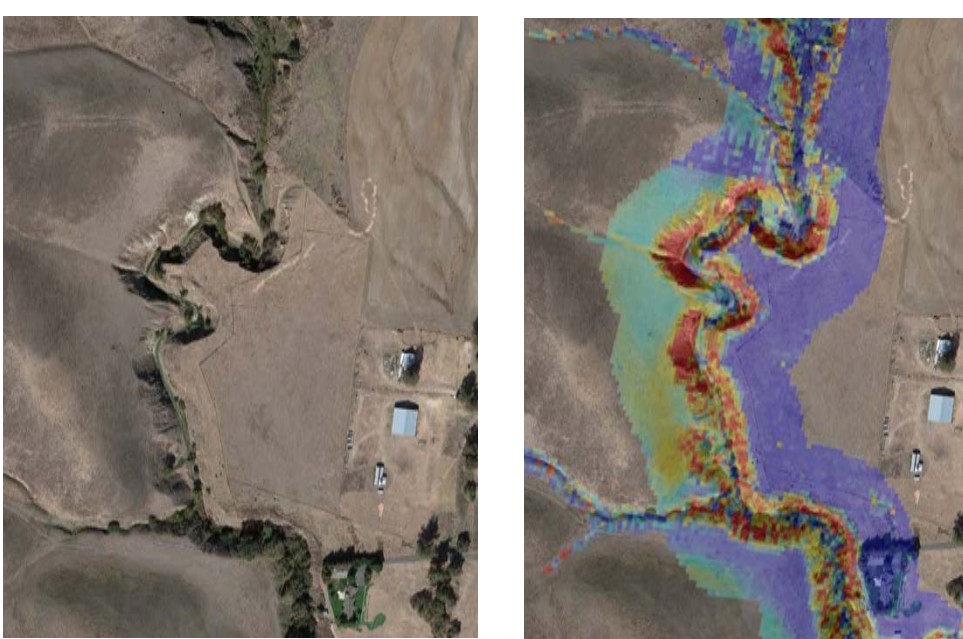

**Figure 5.** Incised channel on lower Tassajara Creek basin showing GEP and estimated specific sediment yield at the reach scale (upper) and air photo (lower left) with pixel scale GEP draped over image (lower right).

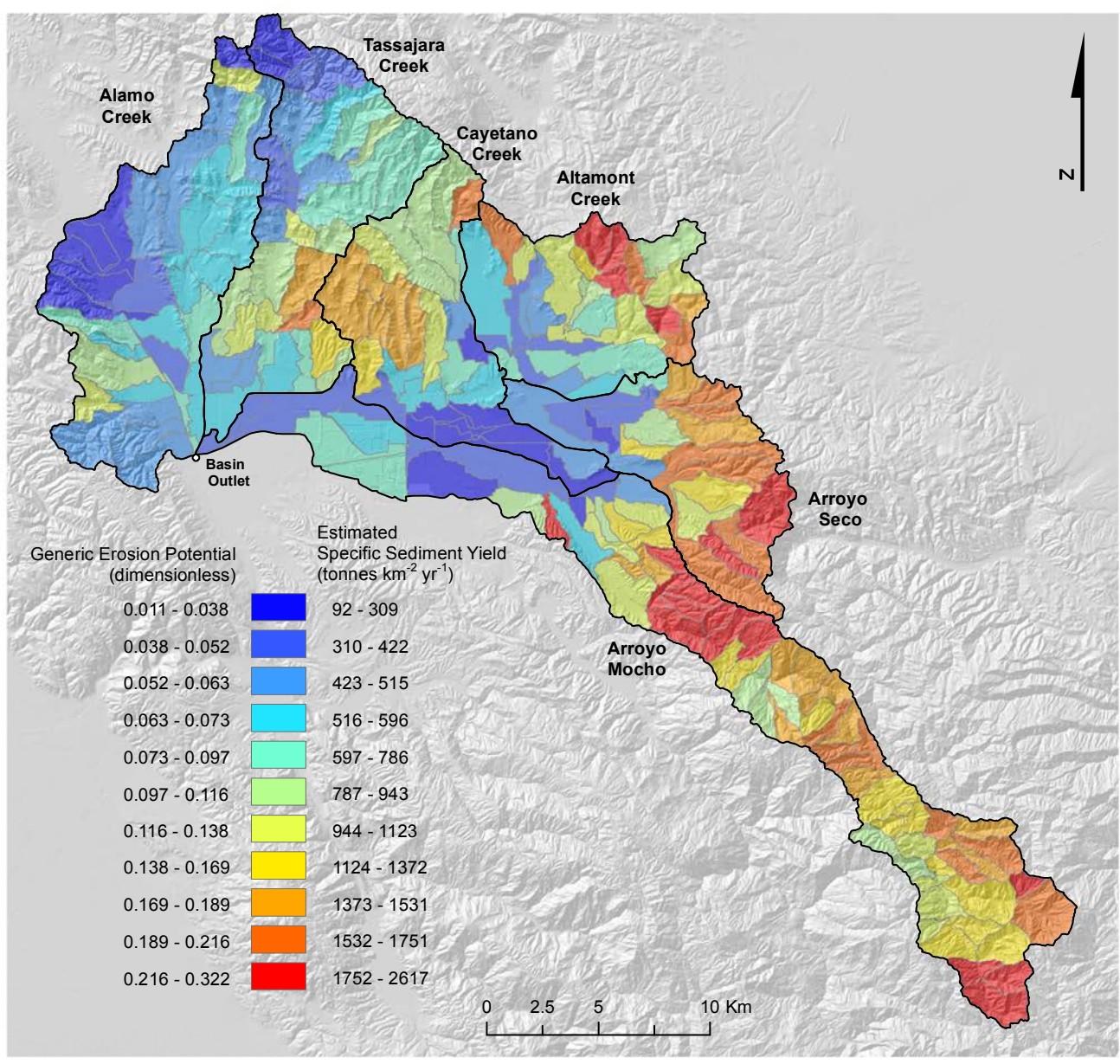

**Figure 6.** GEP and estimated specific sediment yields at the subwatershed scale.

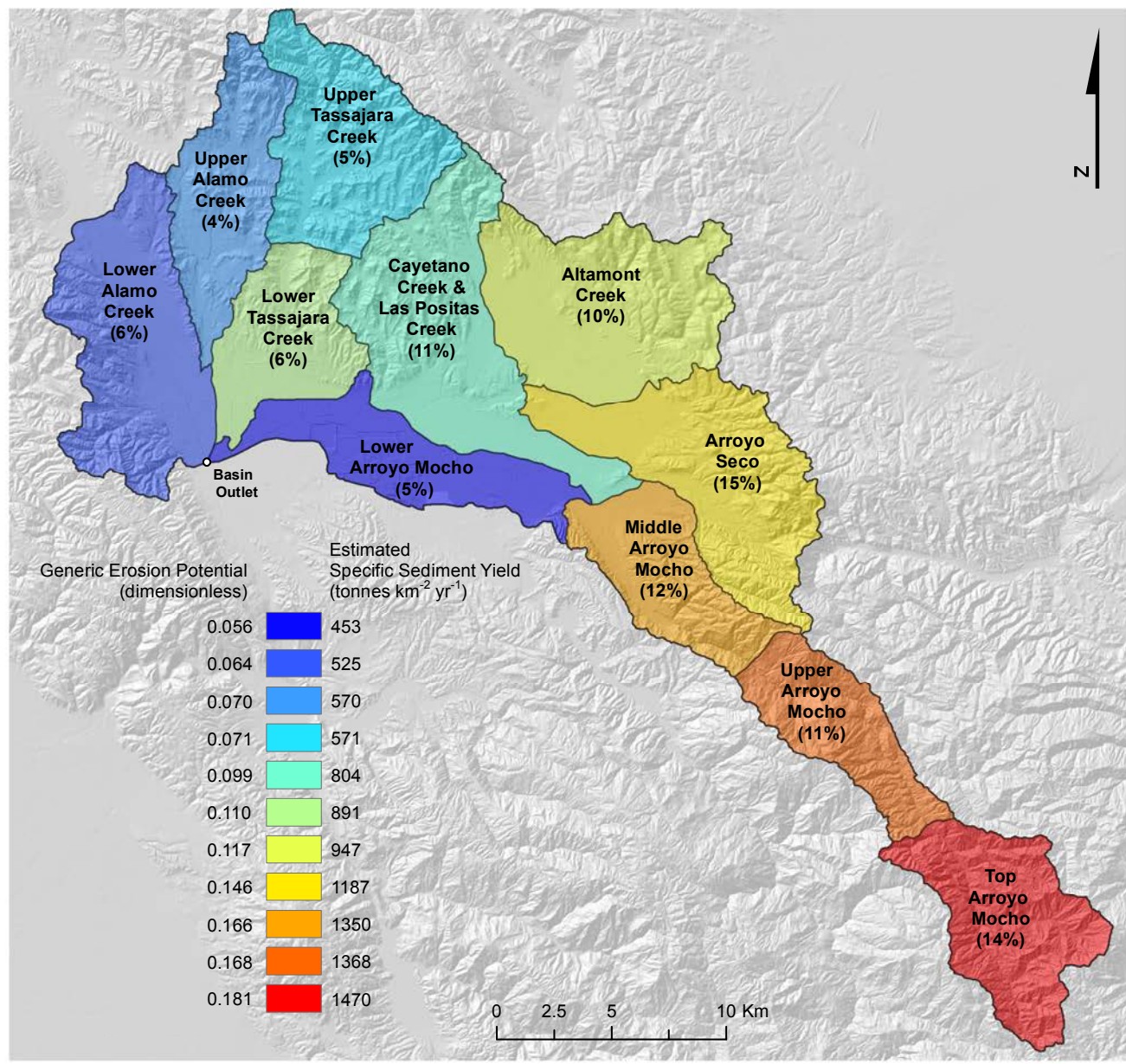

**Figure 7.** GEP and estimated specific sediment yields for the 11 major basins in the Arroyo Mocho watershed. Values in parentheses are the percentage of the total estimated sediment yield for Arroyo Mocho contributed by each basin.

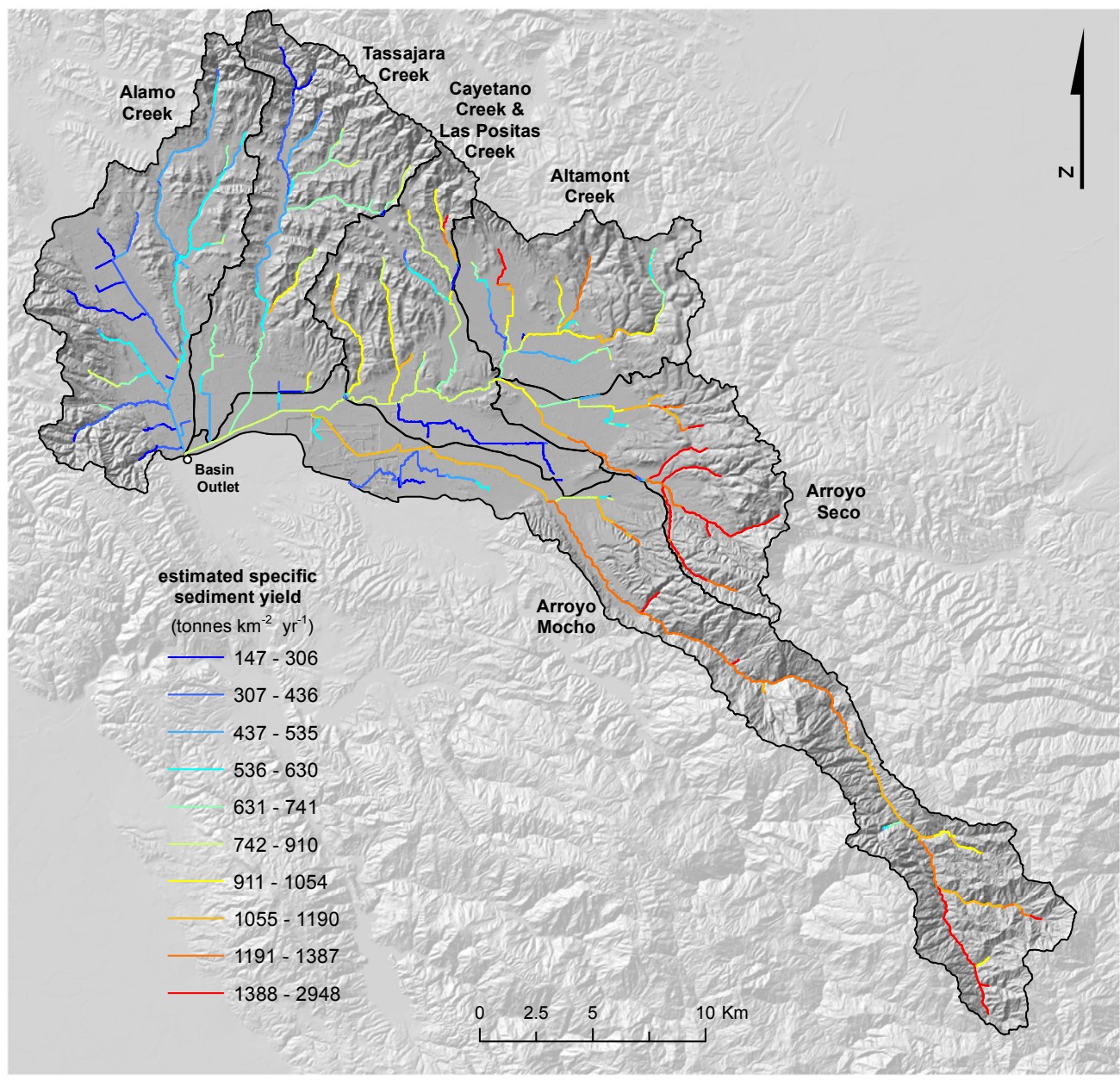

**Figure 8.** Estimated specific sediment yield aggregated down through the stream network for mainstem streams (draining areas > 2 km$^2$).

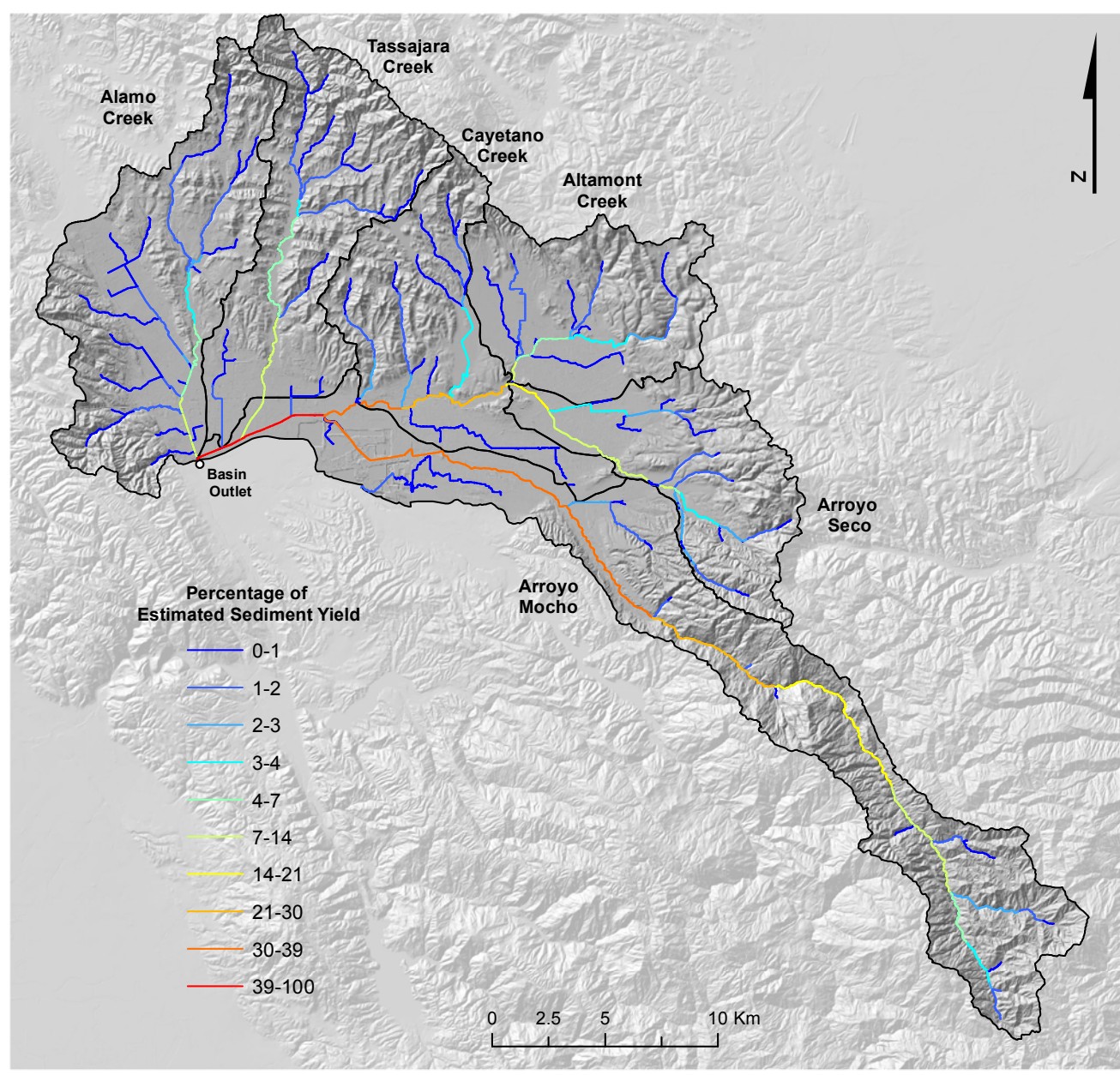

**Figure 9.** Percentage of total estimated sediment yield aggregated downstream through the stream network for mainstem streams (draining areas > 2 km$^2$).

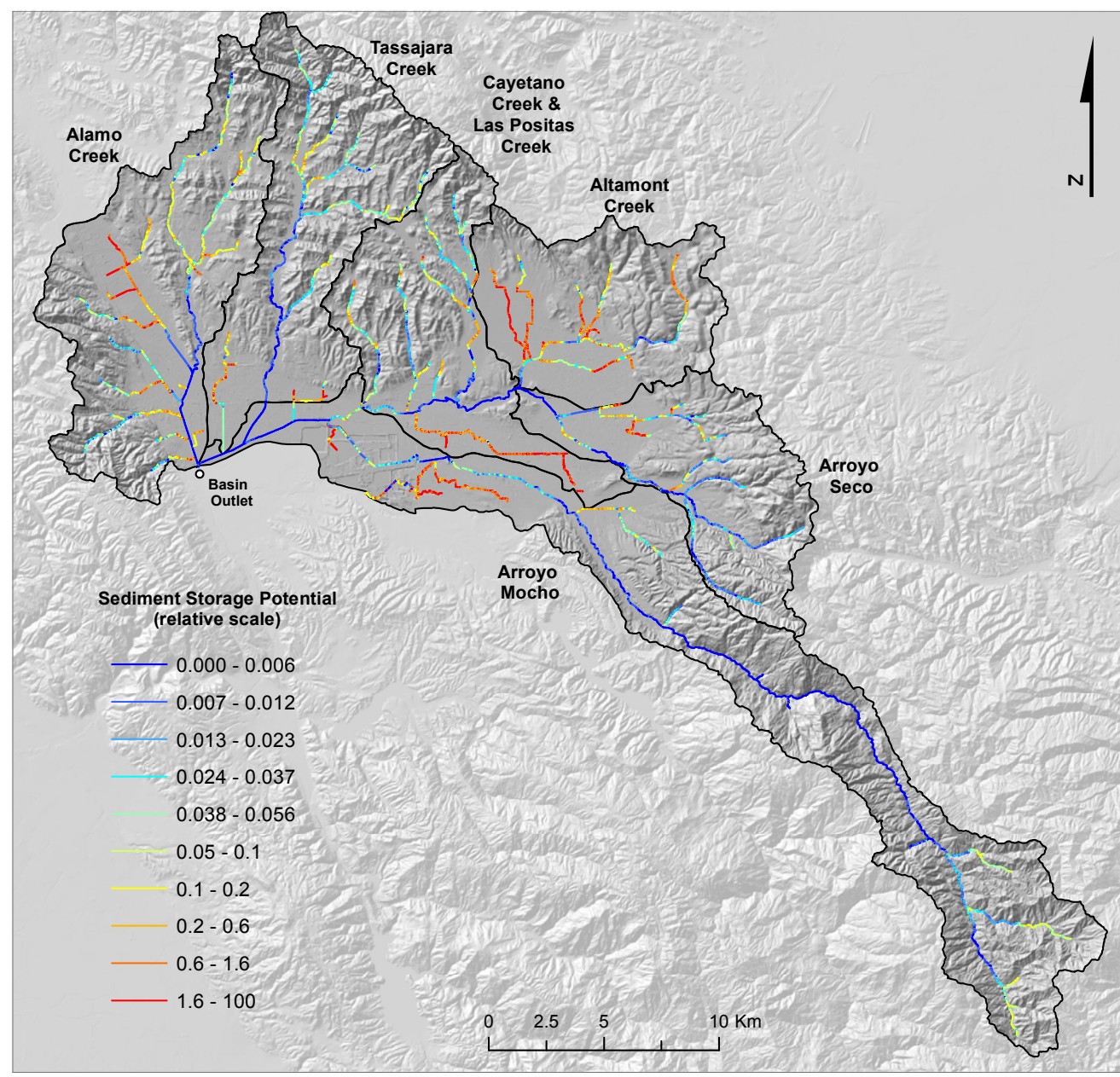

**Figure 10.** Estimated relative sediment storage potential for mainstem channels (draining areas > 2 km$^2$) based on valley width index and stream power (see text).

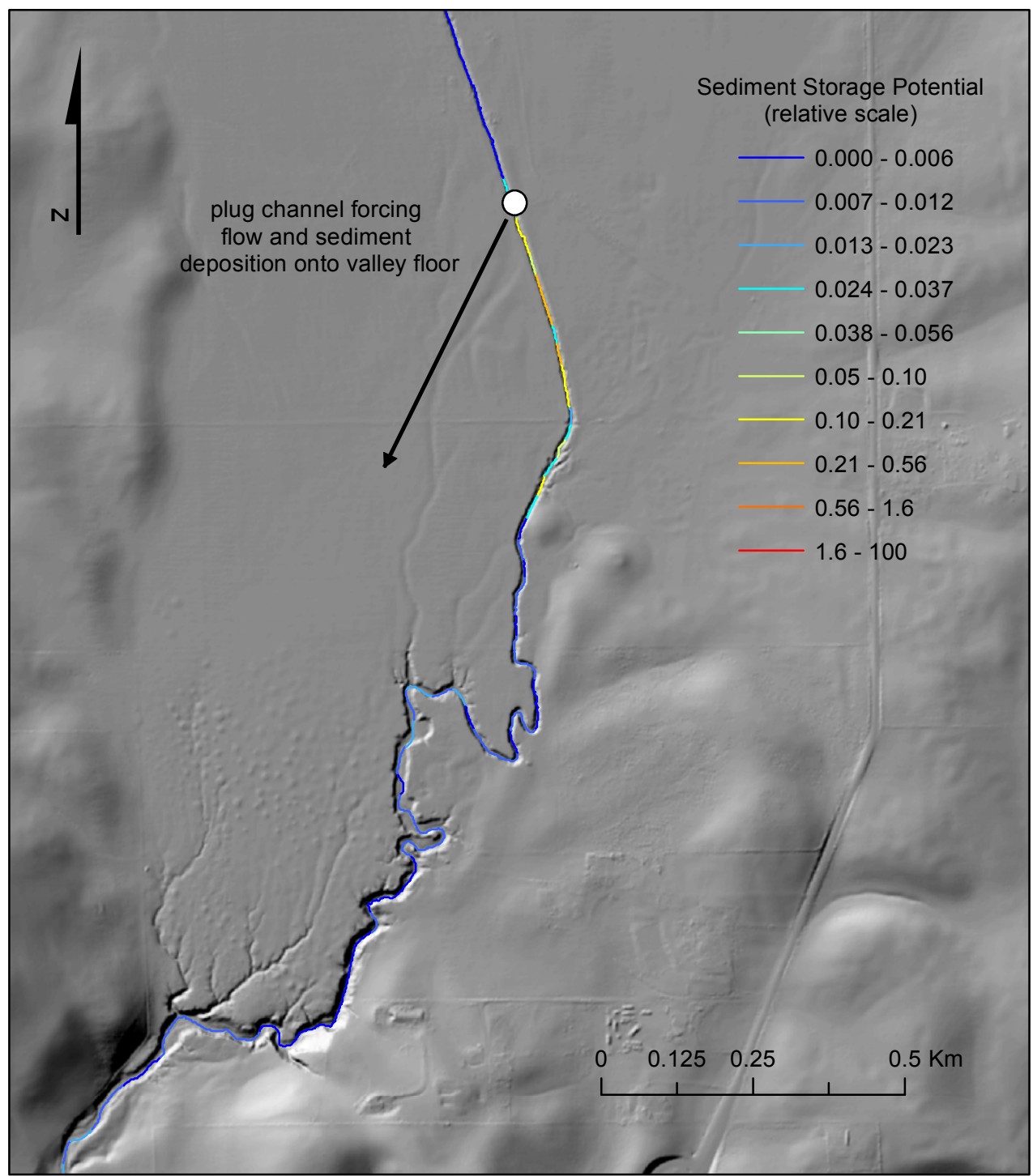

**Figure 11.** Example in the Cayetano Creek tributary showing how sediment storage potential can help identify locations to reconnect channels to floodplains.