# Peer review of "Delineating incised stream sediment sources within a San Francisco Bay tributary basin"

_Earth Surface Dynamics, 2016_

## Referee Comment (RC1)

The authors present a set of GIS tools for the analysis of erosion and sediment yield at river basin scale and propose an application to a 573 km2 watershed in California. The tools developed are easily applicable and could help watershed managers in objectively prioritizing sediment sources and recognizing areas for sediment storage.

I wish to recommend some refinements in the introduction and in the description of methods

**Section 1: Introduction and Objective**

Sediment trapping by dams and gravel mining could be mentioned amongst the processes that cause channel incision (Kondolf, 1997; Surian and Rinaldi, 2003).

More papers (e.g., Brasington et al., 2000, Fuller et al., 2003; Picco et al., 2013) on the assessment of channel erosion and aggradation by means of DEM differencing could be cited, including the work by Wheaton et al. (2010), which provided an important advance in uncertainty assessment.

**Section 3: Methods**

Line 23 of page 4 enounces that Eq. 1 employs slope gradient and local topographic convergence: the variable *S*, which apparently corresponds to slope, could be clearly defined. More details could be provided on the variation of the parameter *b* with topographic convergence.

Page 5, lines 24-25: "Eq. 2 simulates that grasses are less effective..." This is a rather strong simplification of the complex interactions between vegetation cover and erosion. What evidences support this ranking in the effectiveness of vegetation types in the studied area? The authors could consider the relations of their approach with methods commonly used to evaluate the influence of vegetation on erosion, such as USLE – RUSLE C-factor.

Equation 3: it can be noted that the conversion factor is equivalent sediment delivery ratio (Walling, 1983), although GEP is an index of erosion, not a quantitative measure of gross erosion.

Sediment yield is a measure of sediment output from a watershed: it incorporates erosion and within watershed redeposition of eroded material. It is a measure lumped at watershed scale, and is dependent on watershed size. Transposing to other sites within the catchment the value of sediment yield computed at the catchment outlet, or ascribing it to land portions, as it has been done in this paper, is thus questionable. Such a transposition could be legitimate provided that the variable is considered an index and not a direct quantification of local sediment yield.

Sediment yields reported for different sectors of the studied watersheds (e.g., Figs. 4 and 5) should be presented as values of an index that enables framing the local susceptibility to erosion in the context of sediment yield assessed at the river basin scale.

**Section 3.6**

The small DEM cell size requires to provide some technical details on the assessment of drainage area. Channel width in most cross sections is presumably much greater than DEM cell size (2 m). If

applied to single DEM cells, the algorithms for upstream watershed extraction and drainage area assessment would fail to identify the whole upstream area. Some information on how this issue was managed could be provided.

**References**

- Brasington, J., Rumsby, B.T., McVey, R.A., 2000. Monitoring and modelling morphological change in a braided gravel-bed river using high resolution GPS-based survey. Earth Surface Processes and Landforms 25, 973–990.
- Fuller, I.C., Large, A.R.G., Milan, D.J., 2003. Quantifying channel development and sediment transfer following chute cutoff in a wandering gravel-bed river, Geomorphology, 54, 307-323.
- Kondolf, G.M., 1997. Hungry Water: Effects of Dams and Gravel Mining on River Channels. Environmental Management, 21, 533–551.
- Picco, L., Mao, L., Cavalli, M., Buzzi, E., Rainato, R., Lenzi, M.A., 2013. Evaluating short-term morphological changes in a gravel-bed braided river using terrestrial laser scanner. Geomorphology, 201, 323-334, DOI: 10.1016/j.geomorph.2013.07.007
- Nicola Surian, Massimo Rinaldi, 2003. Morphological response to river engineering and management in alluvial channels in Italy, Geomorphology, 50, 307-326.

Walling, D.E., 1983. The sediment delivery problem. Journal of Hydrology, 65, 209-237.

Wheaton, J.M., Brasington, J., Darby, S.E., Sear, D.A., 2010. Accounting for uncertainty in DEMs from repeat topographic surveys: improved sediment budgets. Earth Surf. Process. Landf., 35, 136–156, doi:10.1002/esp.1886

---

## Referee Comment (RC3)

**Manuscript No.:**  esp-2016-5

**Manuscript Title:**  Delineating incised stream sources within a San Francisco Bay tributary basin.

**Overview:**  The authors use geospatial analysis to identify "hotspots" of erosion and sediment delivery in an incised channel network within a tributary to the San Francisco Bay in California, USA.  Channel adjacent sediment sources are identified and quantified based on a combination of DEM-derived slope gradient and morphology, as well as the density and size of riparian vegetation.  Rates of sediment delivery are constrained by measured sediment yield data for the watershed.  Rates are summarized at the 30-m reach scale and the subbasin scale to demonstrate spatially explicit sediment delivery within the larger watershed.

**General Comments:**  I agree with anonymous referee #2 that this is a relevant and interesting study, which can provide restoration practitioners with some relatively simple tools to help identify and prioritize restoration activities in a cost-effective manner.  I also agree that more information is needed on the assumptions and methods used in the analysis.  Overall, I recommend this paper to be approved if additional detail is added to the manuscript.

**Specify and Discuss Hydraulic Geometry Relationships for Modeled Watershed -** The authors cite Dunne and Leopold (1978) for hydraulic geometry relationships related to predicting bankfull width and depth.  I suggest that the authors state both equations in the manuscript.  Also, given that Dunne and Leopold (1978) only published the regression lines and not the data used to derive the relationships, I wonder how accurate these regional regressions are for predicting bankfull width and depth for the modeled watersheds.  This is important, since the channel buffer width (i.e., source areas) is dependent on the bankfull channel width.   As such, the authors should briefly describe whether the hydraulic geometry relationships from Dunne and Leopold (1978) predicted reasonable values for bankfull width and bankfull depth for the modeled watershed(s).

**More Detail Needed on the Channel Buffering Technique -** The manuscript cites Perroy et al. (2010) as the rationale for using the buffer of 6 times the bankfull width. The Perroy paper mentions that their buffering technique scaled with stream order, but did not discuss a specific factor of 6.  More information is needed on how the authors came up with the value of 6 times the bankfull width to determine their channel buffer width.  Ideally, the width of the buffer should relate to the height of the bank and the geotechnical properties of the bank material, as these properties will largely dictate the extent of the adjacent hillslope subject to failure.

**More Detail Needed on How Erosion Reduction Equation was Derived –** There needs to be more detail on the how the curve (Figure 2) and the equation (Equation 2) were created.

**Erosion Source Areas and Proximity to Channel Knickpoints** – This might be beyond the scope of this paper, but it would be interesting to see if the erosion patterns were related to the location of discontinuities in the channel profile. Some recent literature has suggested that inner gorge failure (i.e., bank failure) happens pervasively downstream of channel knickpoints/knickzones (Bennett et al., 2016). This could give restoration practitioners even more insight on where to concentrate their restoration activities.

**References:**

Bennett, G.L., S.R. Miller, J.J. Roering, and D.A. Schmidt. 2016. Landslides, threshold slopes, and the survival of relict terrain in the wake of the Mendocino Triple Junction. Geology. doi:10.1130/G37530.1.

---

## Referee Comment (RC2) · Anonymous Referee #2 · 14 Mar 2016

**1   General Comments**

This is an original and interesting study. The well-structured and well-written paper describes an empirical approach of assessing (potential) sediment yield using topographic and vegetation information from airborne LiDAR data. Slope, convergence and the local (!) contributing area are combined for each raster cell into a 'generic erosion potential' index. The analysis is restricted to a buffer zone around the channel network within a c. 500 sqkm catchment where incision and bank failure and erosion present issues for sediment management. Measured mean sediment yield is used to 'translate' the GEP index into estimated sediment yield that is reduced based on an empirical relationship of vegetation coverage and yield reduction; vegetation coverage is assessed using first vs last pulse returns of airborne LiDAR data. The authors use

the pixel-based data for upscaling to various scales, e.g. channel reaches and sub-catchments. My biggest concern is the fact that the contributing area beyond the 3x3 neighbourhood of each raster cell, and potential decoupling of sediment sources and the channel network (and also within or along the channel network) are apparently not accounted for. A more thorough and reproducible description of some of the methods used would help resolve possible misunderstandings on my part. In all, I think that moderate revisions will make this contribution an interesting paper, not only for the scientific community but especially for practicioners.

**2 Specific Comments**

- p1 l23: common erosional cycle suggest that this is a totally natural, authigenic phenomenon; however, as you elaborate in the following lines, this kind of ero-sional phenomenon represents the response to environmental or anthropogenic changes. I'd like to suggest you emphasize more the 'reaction to change' aspect; if you insist on the 'common erosional cycle', please give a reference

- p2 l14: with 'shallow failures' you mean bank failures ?

- p2 l16: The virtual watershed system or software is only named and explained in the methods section (p3 l 36); here, the reader might wonder if it is a purely virtual experiment , not a study using data from a real landscape. You also mention a 'synthetic river network' the reader might possibly misinterpret along the same lines. I suggest you shift the short explanation of the virtual watershed system from the methods section to the introduction.

- p3 l8f: if you have aggrading channel reaches in your area, then the approach of dividing the outlet sediment yield by the size of the incised area+buffer is probably affected by this: The sum of sediment delivery from the channel banks could be

larger than the measured sediment yield due to intermediate sedimentation. I feel you should discuss this where you explain the transfer from the GEP to sediment delivery

- p3 l28: A figure with one or more 'typical' cross-sections through the valley, maybe one in the lower and one in the upper reaches, could help visualise and justify your statement that the valley floors and hillslopes are not eroded, and the choice of your buffer width. Moreover, I could only guess that the 6times bankful width refers to the total buffer width and not its radius to both sides.)

- section 3.1: the description of how you produced the channel network is too vague, it only refers to the literature; please add at least a brief description of the (major) steps involved. E.g., how did you identify the channel initiation points for the 'synthetic' network ? Another weakness here is that the reader does not know how you delineated the segments: manually ? automatically ? On what basis or using which criteria ?

- section 3.2: l27ff lack the description of S and its unit (percent ? degree ?). You only give a reference for the derivation of b, I feel that you need to describe the major steps at least. I cannot figure out how GEP could be constrained to values between 0 and 1 as long as I do not know the range of S and b. For example, if the slope on a raster cell is steep, and if it has, say, 5 contributing neighbours, the product of S and aL is supposed to be quite large; if you divide it by b which is only said to be 'around 1', GEP will be larger than 0, right ? Moreover, the contributing area of a raster cell can be MUCH larger than just the up to 8 surrounding pixels, what about the mouth of a small tributary rill cutting the riverbank ? Above, I understood that you're linking the contributing areas to the channel network using flow accumulation (and direction) within the buffer zone; and I do not think you can exclude the contributing area of a cell when assessing erosion potential (c.f. stream power index)... Moreover: Would you agree that a

pit with 8 contributing neighbours is prone to erosion ? Well, the slope is 0 inside a pit, but depending on the slope derivation algorithm this is not always the case. It may be that I misunderstood parts of the description, but I feel it has to be more extensive in order to be reproducible; the reader shouldn't have to read Miller and Burnett (2007) first.

- section 3.3: I can understand the need to include a parameterisation for vegetation effects, and deriving this using the LiDAR first+last signal approach is fine for me. However, I think that how you arrive at the erosion reduction factor is poorly describe. You mention 'supporting literature' and 'own observations', but it is not clear how you got the percentage reductions used in Fig 2 to derive equation 2. In line 26 I can only guess why the vegetation height derived from first+last pulse should also represent vegetation density; please write a sentence to explain. Line 31: In my opinion you do not normalize the vegetation height, but you apply equation 2 to the 'elevation height' raster.

- section 3.4: As stated earlier, I feel you should be (more) explicit about some assumptions: (i) within the buffer zone, no intermediary barriers or buffers exist that decouple sediment sources from the channels (ii) there is no intermediary deposition or aggradation on the longer timescale within the whole channel network. This (at least) has to be assumed, because the measured sediment yield at the output is related to the entire size of the buffer zone. However, I agree that such simplifications have to be implemented, especially when the aim of setting up a practicioner-friendly tool. Moreover: Please reconsider if a linear relationship of GEP and sediment yield is appropriate. Statistical relationships contributing area and discharge or sediment yield have a sort of power law...

- section 3.5: In lines 19 and 23 (also Fig. 4, 5 and 6), I think that what you're computing is the specific sediment yield, because you divide the sediment yield by the upstream drainage area. Furthermore, please be explicit here if you divide

by the total upstream drainage area or only by the 'buffered' part of it. For the channel segments (Fig. 6), would it make sense to report and discuss sediment yield instead of specific sediment yield ?

- section 3.6: Similarly to the vegetation height index, you could report here the range of the storage potential index within the study area

- section 4.1: p7l10 suggests that you selected c. 50 percent of the catchment area for visual qualitative validation. How did you infer 'much more stable banks' from satellite imagery, except by the presence of vegetation ? Strictly speaking: As vegetation is detected by LiDAR data analysis and included in your index, you simply observe the effect of the index computation (what would be a verification of your approach) rather than validating the GEP index by observing eroded vs. stable banks. You could shortly discuss here a validation in similar, perhaps neighbouring catchments with available sediment measurements to check the transferability of your index at least for comparable catchments.

- section 4.2: A question to discuss: A patch of high GEP-pixels would only be contributing if there is a continous sediment pathway towards the channel. Can you exclude that there are intermediary flat areas or other buffers that decouple the sediment source (which your index may correctly identify) and the channel ? How do you deal with the channels that you know are aggrading ? I like the discussion of average yield vs episodic character of erosion.

- section 4.3: Check if 'summed and area weighted' sediment yields could not be better termed 'specific sediment yields'. In l8, I suggest you replace 'total load' by 'total sediment yield at the outlet'.

- section 4.4: This section is particularly important for practicioners who want to implement the approach. In the first subsection, please add a sentence on whether and how also the 'raw' i.e. pixel-scale index could be useful once a river reach or

subwatershed has been identified as priority. The second subsection is too short I think, because the results of the storage index are not presented in any detail; you could, for example, highlight one or two parts of the catchment where the storage potential is particularly high or low, and explain why. Moreover, the question pops up how the floodplains that should be reconnected are disconnected at the time being; you could include this information in the study area section, for example, or where you explain the storage potential index. What measures would have to be taken to reconnect floodplains to (heavily incised !) channels ?

- chapter 5: Personally, I would prefer if the two proposals for potential improvements were part of the discussion section, with a (slightly longer) proper conclusion chapter 5. Regarding the second improvement: Can one really think of a (even average) ratio bedload vs suspended load, given different forcing magnitudes, and also given the episodic character of events in the study area that has already been mentioned ? Does the ratio of fine and coarse sediment within the sediment sources equal the ratio of fluvial sediment in transport or at the outlet ? References to that end ?

- p9 l34f: The reviewer explicitly joins the authors in this statement of gratitude ;-)

**3  Technical Corrections**

- p1 l25: 'increase transport capacity relative to...' instead of 'increase transport relative to...'

- p2 l2: delete 'could be applied'

- p2 l14: (GEP=Generic Erosion Potential)

- p2 l16: downstream

- p3 l5: either 'sediment...is now supplied' or 'sediments...are now supplied'

- p5 14: Split the sentence here: 'or infer generalized relationships. For example, Pelletier...'

- p6 l19: write 'outlet' rather than 'bottom'

- p6 l20: might be picky, and might be due to me not being a native speaker: Sediment is not eroding, it is being transported or transferred to a channel reach, after having been eroded or mobilised.

---

## Author Comment (AC1) · 14 May 2016

**Authors' response to referee comments on "Delineating incised stream sediment sources within a San Francisco Bay tributary basin"**

Responses are *italicized*

**Referee 1: Lorenzo Marchi**

**General comments:**

The authors present a set of GIS tools for the analysis of erosion and sediment yield at river basin scale and propose an application to a 573 km2 watershed in California. The tools developed are easily applicable and could help watershed managers in objectively prioritizing sediment sources and recognizing areas for sediment storage. I wish to recommend some refinements in the introduction and in the description of methods

We thank the referee for the constructive comments and suggested refinements, we have incorporated them all into the paper.

**Specific comments:**

Section 1: Introduction and Objective

Sediment trapping by dams and gravel mining could be mentioned amongst the processes that cause channel incision (Kondolf, 1997; Surian and Rinaldi, 2003).

**We added this suggestion and references to the introduction.**

More papers (e.g., Brasington et al., 2000, Fuller et al., 2003; Picco et al., 2013) on the assessment of channel erosion and aggradation by means of DEM differencing could be cited, including the work by Wheaton et al. (2010), which provided an important advance in uncertainty assessment.

We added these references as suggested to the introduction.

Section 3: Methods

Line 23 of page 4 enounces that Eq. 1 employs slope gradient and local topographic convergence: the variable S, which apparently corresponds to slope, could be clearly defined. More details could be provided on the variation of the parameter b with topographic convergence.

*We have added more details to the methods here to clarify slope and variation of b, the text now reads:*

*"GEP (Eq. 2) is calculated from topographic attributes of slope gradient and topographic convergence (planform curvature) derived from the DEM:*

 $GEP = S \cdot a_L/b$

(2)

where GEP is the generic erosion potential, S is slope gradient (m/m),  $a_L$  is the local contributing area to a DEM pixel, and b provides an estimate of the total contour length crossed by flow exiting a pixel. Local contributing area  $a_L$  is calculated using the D-infinity flow direction algorithm (Tarboton 1997), which allows downslope dispersion. The flow direction for each pixel is calculated using each of eight triangular facets defined by a DEM grid point and the eight adjacent points, where values range from 0 - 8. For each facet having flow out of the pixel, we use the projection of flow direction on the exterior facet edge as a measure of contour length crossed by flow exiting the pixel from that facet. These projection lengths are summed over all edges with outgoing flow to provide an estimate of contour length b for flow exiting the pixel, where values range from 0 - 4. Contour length for planar flow is one pixel width, for divergent flow it is greater than one pixel width, and for convergent flow it is less than one pixel width. The ratio  $a_L/b$  therefore incorporates effects of topographic convergence (Miller and Burnett 2007).

Page 5, lines 24-25: Eq. 2 simulates that grasses are less effective..." This is a rather strong simplification of the complex interactions between vegetation cover and erosion. What evidences support this ranking in the effectiveness of vegetation types in the studied area? The authors could consider the relations of their approach with methods commonly used to evaluate the influence of vegetation on erosion, such as USLE or RUSLE C-factor.

**We have removed the line in question, and explicitly mention the RUSLE C-factor here in comparison to our approach as suggested.**

Equation 3: it can be noted that the conversion factor is equivalent sediment delivery ratio (Walling, 1983), although GEP is an index of erosion, not a quantitative measure of gross erosion.

Sediment yield is a measure of sediment output from a watershed: it incorporates erosion and within watershed redeposition of eroded material. It is a measure lumped at watershed scale, and is dependent on watershed size. Transposing to other sites within the catchment the value of sediment yield computed at the catchment outlet, or ascribing it to land portions, as it has been done in this paper, is thus questionable. Such a transposition could be legitimate provided that the variable is considered an index and not a direct quantification of local sediment yield. Sediment yields reported for different sectors of the studied watersheds (e.g., Figs. 4 and 5) should be presented as values of an index that enables framing the local susceptibility to erosion in the context of sediment yield assessed at the river basin scale.

We have added to the description of the conversion factor does not account for any deposition within the channel, and therefore underestimates sediment sediment supply, but likely corresponds to the correct order of magnitude in the context of sediment budgeting technology (Reid and Dunne 1996).

**Section 3.6**

The small DEM cell size requires to provide some technical details on the assessment of drainage area. Channel width in most cross sections is presumably much greater than DEM cell size (2 m). If applied to single DEM cells, the algorithms for upstream watershed extraction and

drainage area assessment would fail to identify the whole upstream area. Some information on how this issue was managed could be provided.

Channel width is not calculated from the DEM, rather we use a regional regression based on drainage area to estimate channel width. Drainage area is calculated for each grid cell based on the all the upstream grid cells flowing into that cell. The application here is not to a single grid cell, but to a stream reach, that ranges in length from 2 - 80 m, and contains the attributes for drainage area, valley width, channel width, gradient, etc. in order to make the calculation for the sediment storage potential index. We have clarified this in the text adding "For all reaches in the stream network, gradient, drainage area, and valley width were extracted from the DEM using algorithms within NetMap (Miller et al. 2002, Benda et al. 2007)."

**References**

Brasington, J., Rumsby, B.T., McVey, R.A., 2000. Monitoring and modelling morphological change in a braided gravel-bed river using high resolution GPS-based survey. Earth Surface Processes and Landforms 25, 973 990.

Fuller, I.C., Large, A.R.G., Milan, D.J., 2003. Quantifying channel development and sediment transfer following chute cutoff in a wandering gravel-bed river, Geomorphology, 54, 307-323. Kondolf, G.M., 1997. Hungry Water: Effects of Dams and Gravel Mining on River Channels. Environmental Management, 21, 533II551.

Picco, L., Mao, L., Cavalli, M., Buzzi, E., Rainato, R., Lenzi, M.A., 2013. Evaluating short-term morphological changes in a gravel-bed braided river using terrestrial laser scanner.

Geomorphology, 201, 323-334, DOI: 10.1016/j.geomorph.2013.07.007

Nicola Surian, Massimo Rinaldi, 2003. Morphological response to river engineering and management in alluvial channels in Italy, Geomorphology, 50, 307-326.

Walling, D.E., 1983. The sediment delivery problem. Journal of Hydrology, 65, 209-237. Wheaton, J.M., Brasington, J., Darby, S.E., Sear, D.A., 2010. Accounting for uncertainty in DEMs from repeat topographic surveys: improved sediment budgets. Earth Surf. Process. Landf., 35, 136III, 156, doi:10.1002/es

**Referee 2: anonymous**

**General Comments:**

This is an original and interesting study. The well-structured and well-written paper describes an empirical approach of assessing (potential) sediment yield using topographic and vegetation information from airborne LiDAR data. Slope, convergence and the local (!) contributing area are combined for each raster cell into a 'generic erosion potential' index. The analysis is restricted to a buffer zone around the channel network within a c. 500 sqkm catchment where incision and bank failure and erosion present issues for sediment management. Measured mean sediment yield is used to 'translate' the GEP index into estimated sediment yield that is reduced based on an empirical relationship of vegetation coverage and yield reduction; vegetation coverage is assessed using first vs last pulse returns of airborne LiDAR data. The authors use the pixel-based data for upscaling to various scales, e.g. channel reaches and subcatchments. My biggest concern is the fact that the contributing area beyond the 3x3 neighbourhood of each raster cell, and potential decoupling of sediment sources and the channel network (and also within or along the channel network) are apparently not accounted for. A more thorough and reproducible description of some of the methods used would help resolve possible misunderstandings on my part. In all, I think that moderate revisions will make this contribution an interesting paper, not only for the scientific community but especially for practicioners.

We wish to thank the referee for their particularly thoughtful and constructive review and comments. Concerns with the methods are due primarily to our lack of detail that has now been added, and is also addressed in the referee's specific comments later. We have incorporated most of the suggestions and they have greatly improved the paper. We address the referee's biggest concerns here too, in three parts:

1. To clarify that generic erosion potential (GEP) is not used to characterize active incision/downcutting of channels (which would require a different method including the entire upslope drainage area, e.g. stream power), we have added the following text in the GEP methods section: "We specifically use GEP as a relative index of erosion potential for shallow failures along steep banks of incised channels in the Arroyo Mocho study area. GEP is not used to estimate erosion potential for downcutting the channel bed or bank erosion of outside river bends. As mentioned earlier, the incised channels in the Arroyo Mocho watershed do not appear to be actively downcutting, rather sediment is now supplied to most channels by shallow slides and failures of oversteepened banks (channel widening), a common progression in the cycle of incised channels following downcutting (Schumm et al. 1984)."

2. To address the referee's concern of using local contributing area for erosion potential, we have added text to the GEP methods explaining: "While upslope drainage area is often used as a surrogate for subsurface flow in some erosion models, most locations on the hillslope receive contributions from a small proportion of the upslope contributing area due to the low velocity of subsurface flow (Barling et al., 1994, Beven and Freer 2001, Borga et al. 2002). We use local contributing area for erosion potential rather than the entire upslope drainage area because (1) conceptually, pore pressures measured during storms in shallow soils correlate poorly with topography (e.g., Dhakal and Sullivan 2014), but convergent areas tend to exhibit persistently high water content, deeper soils, and are highly responsive to rainfall, and (2) empirically, local

contributing area has been shown to better predict shallow failures than total contributing area in several studies (Miller 2004, Miller and Burnett 2007)." While it seems reasonable that the degree of antecedent soil moisture may correlate with total contributing area (e.g., Hotta et al., 2010), the time scales for transmission of pressure changes in response to rainfall can span from minutes to centuries (Iverson 2000), making it difficult to discern how much of the contributing area influences pore-pressure response at any particular point during any particular storm. Intense rainfall tends to be of short duration, and shallow slides tend to be triggered by intense rainfall (with high antecedent conditions). Also, if you apply the kinematic wave approach used for Shalstab-like models, and apply a Darcy velocity for saturated flow, the upslope length of contributing area varies with storm duration.

3. To address the referee's concern that sediment sources decoupled from the channel may not be accounted for, we have added text to the GEP methods section clarifying "Because we confine the analysis to the buffered incised channel network, we assume that all sediment eroded from the inner gorges, arroyos, and gullies is delivered directly to the channel network, as confirmed during two days of field observations across the watershed. When practitioners want to include other forms of erosion further from the channel, the proportion of sediment delivered to the channel should be estimated based on different topographic attributes (e.g. gradient and confinement, Miller and Burnett 2007), which is possible within the NetMap terrain mapping platform (Benda et al. 2007) and other similar approaches." Also, we have added text explicitly recognizing that channel storage/aggradation is not accounted for: "The conversion to sediment yield does not account for any deposition within the channel and therefore underestimates the spatially explicit sediment supply rates, but likely corresponds to the correct order of magnitude within the context of sediment budgeting technology (e.g., Reid and Dunne 1996)." New references cited above:

Barling, D. B., Moore, I. D., and Grayson, R. B.: A quasi-dynamic wetness index for characterising the spatial distribution of zones of surface saturation and soil water content. Water Resour. Res. 30, 1029–1044, 1994.

Beven, K., and Freer, J.: A dynamic TOPMODEL. Hydrol. Process. 15, 1993–2011, 2001.

Borga, M., Dalla Fontana, G., and Cazorzi, F.: Analysis of topographic and climatic control on rainfall-triggered shallow landsliding using a quasi-dynamic wetness index, Journal of Hydrology, 268, 56-71, 2002.

Dhakal, A. S., and Sullivan, K.: Shallow groundwater response to rainfall on a forested headwater catchment in northern coastal California: implications of topography, rainfall, and throughfall intensities on peak pressure head generation, Hydrological Processes, 28, 446-463, 2014.

Hotta, N., Tanaka, N., Sawano, S., Kuraji, K., Shiraki, K., & Suzuki, M. (2010). Changes in groundwater level dynamics after low-impact forest harvesting in steep, small watersheds. Journal of hydrology, 385(1), 120-131.

*Iverson, Richard M. "Landslide triggering by rain infiltration." Water resources research 36.7 (2000): 1897-1910.*

Miller, D. J.: Landslide Hazards in the Stillaguamish basin: a new set of GIS tools, prepared for the Stillaguamish Tribe of Indians Natural Resource Department, 48 pp., 2004.

*Miller, D. J., and Burnett, K. M. Effects of forest cover, topography, and sampling extent on the measured density of shallow, translational landslides, Water Resour. Res., 43, 1–23, 2007.*

**Specific Comments:**

p1 l23: common erosional cycle suggest that this is a totally natural, authigenic phenomenon; however, as you elaborate in the following lines, this kind of erosional phenomenon represents the response to environmental or anthropogenic changes. I'd like to suggest you emphasize more the 'reaction to change' aspect; if you insist on the 'common erosional cycle', please give a reference

Good point. We have clarified that channel incision is a common erosional response to natural or anthropogenic forcing.

p2 114: with 'shallow failures' you mean bank failures ?

We have clarified here that we mean shallow failures from planar and convergent slopes. These are inner gorge type failures from over steepened banks, not bank erosion (e.g. outside river bends).

p2 116: The virtual watershed system or software is only named and explained in the methods section (p3 1 36); here, the reader might wonder if it is a purely virtual experiment, not a study using data from a real landscape. You also mention a 'synthetic river network' the reader might possibly misinterpret along the same lines. I suggest you shift the short explanation of the virtual watershed system from the methods section to the introduction.

We removed the mention of virtual watershed and synthetic here to avoid confusion and clarify the NetMap terrain mapping platform was used to generate the various GIS layers.

p3 l8f: if you have aggrading channel reaches in your area, then the approach of dividing the outlet sediment yield by the size of the incised area+buffer is probably affected by this: The sum of sediment delivery from the channel banks could be larger than the measured sediment yield due to intermediate sedimentation. I feel you should discuss this where you explain the transfer from the GEP to sediment delivery

Referee #1 had a similar concern, we have clarified in Section 3.4 Conversion to Sediment Yield, that "The conversion to sediment yield does not account for any deposition within the channel and therefore underestimates the spatially explicit sediment yield rates, but likely corresponds to the correct order of magnitude within the context of sediment budgeting technology (Reid and Dunne 1996)."

p3 l28: A figure with one or more 'typical' cross-sections through the valley, maybe one in the lower and one in the upper reaches, could help visualise and justify your statement that the valley floors and hillslopes are not eroded, and the choice of your buffer width. Moreover, I could only guess that the 6times bankful width refers to the total buffer width and not its radius to both sides.)

**We have added a figure with cross sections as suggested and clarified that the buffer is 6 times bankfull width is the total buffer width.**

section 3.1: the description of how you produced the channel network is too vague, it only refers to the literature; please add at least a brief description of the(major) steps involved. E.g., how did you identify the channel initiation points for the 'synthetic' network ? Another weakness here is that the reader does not know how you delineated the segments: manually ? automatically ? On what basis or using which criteria ?

We have included more detail on the major steps used to derive the channel network and how the channel segments were delineated. This section now reads as follows:

"An attributed digital stream network was derived from the DEM using a series of algorithms described by Miller (2002). The following major steps were followed in the construction of the DEM and extraction of the channel network:

- 1. DEM. The 2 m DEM was compiled and resampled from a 0.3 m DEM for the majority of the watershed within the dominant jurisdictional boundary (Alameda County) and a 3 m DEM for small portions of the watershed that lie in other counties, all derived from LiDAR data collected by the U.S. Geological Survey. We wanted to maintain information from the most precise and accurate data in creation of a single, contiguous DEM, while also avoiding creation of breaks in elevation or derivatives of elevation (gradient, curvature) at seams between the different data sources.
- 2. Topographic attributes. Topographic attributes for network extraction were based on unaltered elevation data, prior to drainage enforcement and hydrologic conditioning. The attributes calculated are surface gradient, plan curvature, and the contour length crossed by flow out of a DEM cell, which is used to calculate specific contributing area.
- 3. Drainage reinforcement. Here, we used digitized polygons of open water so flow lines should run down the center of these polygons, and we defined channel initiation points using criteria based on flow accumulation and surface slope, plan (contour) curvature, and flow length over which these criteria are met (Miller 2002, Miller et al. 2015).
- 4. Hydrologically conditioned DEM. We created a hydrologically conditioned DEM where the flow paths out of all closed depressions are identified using a combination of depression filling (Jenson and Domingue, 1988) and carving (Soille et al., 2003).
- 5. Flow and channels. We calculated flow accumulation to identify all channel initiation

points, and trace all channels. We then applied the D-infinity algorithm (Tarboton, 1997) to calculate flow accumulation values down to channel initiation points. Channels were then traced downstream from these points using D-8 flow directions (Jenson and Domingue, 1988) to preclude dispersion of channelized flow. D-8 flow directions are chosen using a combination of steepest descent and largest plan curvature (Clarke et al., 2008). Using algorithms from Miller (2002), the channel network was divided into homogeneous reaches with end-point positions selected that minimize variance of channel gradient, valley width, and drainage area within a reach, while keeping mean reach lengths less than 100 m; each reach was attributed with various topographic measures, including channel slope, valley width, drainage area, etc..

6. Refining Network. We then smoothed channel traces to provide better-placed channel centerlines and more accurate estimates of channel length and gradient. We also validated the delineated channel network using a combination of local knowledge, field surveys, and high-resolution optical imagery.

**More details on methods used to derive of the channel network from the DEM and algorithms used can be found in Miller et al. (2015), Clarke et al. (2008), and Miller (2002).**

section 3.2: 127 ff lack the description of S and its unit (percent ? degree ?). You only give a reference for the derivation of b, I feel that you need to describe the major steps at least. I cannot figure out how GEP could be constrained to values between 0 and 1 as long as I do not know the range of S and b. For example, if the slope on a raster cell is steep, and if it has, say, 5 contributing neighbours, the product of S and aL is supposed to be quite large; if you divide it by b which is only said to be 'around 1', GEP will be larger than 0, right ? Moreover, the contributing area of a raster cell can be MUCH larger than just the up to 8 surrounding pixels, what about the mouth of a small tributary rill cutting the riverbank ? Above, I understood that you're linking the contributing areas to the channel network using flow accumulation (and direction) within the buffer zone; and I do not think you can exclude the contributing area of a cell when assessing erosion potential (c.f. stream power index)... Moreover: Would you agree that a pit with 8 contributing neighbours is prone to erosion ? Well, the slope is 0 inside a pit, but depending on the slope derivation algorithm this is not always the case. It may be that I misunderstood parts of the description, but I feel it has to be more extensive in order to be reproducible; the reader shouldn't have to read Miller and Burnett (2007) first.

**We have rewritten this section to include more detail and text from Miller and Burnett (2007) and other sources. The lack of detail led to confusion for readers and the section 3.2 now reads:**

"To provide a relative estimate of erosion potential within the buffered incised channel network of the Arroyo Mocho watershed, we used a topographic index called GEP (Miller and Burnett 2007, Benda et al. 2011) that combines slope steepness with slope convergence, recognized topographic indicators of shallow landsliding, gully erosion, and sheetwash (Dietrich and Dunne 1978, Sidle 1987, Montgomery and Dietrich 1994, Miller and Burnett 2007, Parker et al. 2010). Slope steepness is a fundamental control on erosion potential, while convergence causes surface and subsurface flow to become concentrated and contribute to erosion potential. GEP (Eq. 2) is calculated from topographic attributes of slope gradient and topographic convergence (planform curvature) derived from the DEM:

$$GEP = S \cdot a_L/b$$

(2)

where GEP is the generic erosion potential, S is slope gradient (m/m),  $a_L$  is the local contributing area to a DEM pixel, and b provides an estimate of the total contour length crossed by flow exiting a pixel. Local ccontributing area  $a_L$  is calculated using the D-infinity flow direction algorithm (Tarboton 1997), which allows downslope dispersion. The flow direction for each pixel is calculated using each of eight triangular facets defined by a DEM grid point and the eight adjacent points, where values range from 0 - 8. For each facet having flow out of the pixel, we use the projection of flow direction on the exterior facet edge as a measure of contour length crossed by flow exiting the pixel from that facet. These projection lengths are summed over all edges with outgoing flow to provide an estimate of contour length b for flow exiting the pixel, where values range from 0 - 4. Contour length for planar flow is one pixel width, for divergent flow it is greater than one pixel width, and for convergent flow it is less than one pixel width. The ratio  $a_I/b$  therefore incorporates effects of topographic convergence (Miller and Burnett 2007). Multiplying by slope provides a basin-wide measure of erosion potential, similar to a slope area product, but using only the local contributing area. While upslope drainage area is often used as a surrogate for subsurface flow, most locations on the hillslope receive contributions from a small proportion of the upslope contributing area due to the low velocity of subsurface flow (Barling et al., 1994, Beven and Freer 2001, Borga et al. 2002). Local contributing area is used for erosion potential rather than the entire upslope drainage area because (1) conceptually, pore pressures measured during storms in shallow soils correlate poorly with topography (e.g., Dhakal and Sullivan 2014), but convergent areas tend to exhibit persistently high water content, deeper soils, and are highly responsive to rainfall events, and (2) empirically, local contributing area has been shown to better predict shallow failures than total contributing area in several studies (Miller 2004, Miller and Burnett 2007). Thus GEP is a dimensionless relative index of erosion potential with most values within a watershed ranging from 0-1, where larger values correspond to steeper, more convergent topography prone to higher landslide densities, surface erosion, and higher gully-initiation-point densities (Miller and Burnett 2007).

We specifically use GEP as a relative index of erosion potential for shallow failures along steep banks of incised channels in the Arroyo Mocho study area. GEP is not used to estimate erosion potential for downcutting the channel bed or bank erosion of outside river bends. As mentioned earlier, the incised channels in the Arroyo Mocho watershed do not appear to be actively downcutting, rather sediment is now supplied to most channels by shallow slides and failures of oversteepened banks (channel widening), a common progression in the cycle of incised channels following downcutting (Schumm et al. 1984)....."

section 3.3: I can understand the need to include a parameterisation for vegetation effects, and deriving this using the LiDAR first+last signal approach is fine for me. However, I think that how you arrive at the erosion reduction factor is poorly describe. You mention 'supporting literature' and 'own observations', but it is not clear how you got the percentage reductions used in Fig 2 to derive equation 2. In line 26 I can only guess why the vegetation height derived from first+last pulse should also represent vegetation density; please write a sentence to explain. Line

31: In my opinion you do not normalize the vegetation height, but you apply equation 2 to the 'elevation height' raster.

We have removed any reference to vegetation density and we've added text in this section, clarifying: "Based on one day of field observations across the watershed where the vegetation height and associated erosion reduction (root spread) were visually estimated, we plotted and derived the best fit equation (Eq. 3, Fig. 2) that governs how erosion potential is reduced by vegetation height and use that to scale the GEP index as described below.

 $Erosion \ reduction = 0.1906 \ ln(vegetation \ height \ in \ m) + 0.136$ (3)

To reduce GEP based on this relationship, we first created a vegetation height grid (2 m) in GIS using the first (representing the tallest vegetation) and last (representing the ground surface) return LiDAR points. We then created an erosion reduction grid by multiplying the vegetation height grid by Eq. 3.

section 3.4: As stated earlier, I feel you should be (more) explicit about some assumptions: (i) within the buffer zone, no intermediary barriers or buffers exist that decouple sediment sources from the channels (ii) there is no intermediary deposition or aggradation on the longer timescale within the whole channel network. This (at least) has to be assumed, because the measured sediment yield at the output is related to the entire size of the buffer zone. However, I agree that such simplifications have to be implemented, especially when the aim of setting up a practicioner-friendly tool. Moreover: Please reconsider if a linear relationship of GEP and sediment yield is appropriate. Statistical relationships contributing area and discharge or sediment yield have a sort of power law...

We have added text to Section 3.2, clarifying: "Because we confine the analysis to the buffered incised channel network, we assume that all sediment eroded from the inner gorges, arroyos, and gullies is delivered directly to the channel network, as confirmed during two days of field observations across the watershed. When practitioners want to include other forms of erosion further from the channel, the proportion of sediment delivered to the channel should be estimated based on different topographic attributes (e.g. gradient and confinement, Miller and Burnett 2007), which is possible within the NetMap terrain mapping platform (Benda et al. 2007) and other similar approaches."

We have also added text in Section 3.4 Conversion to Sediment Yield, clarifying that "The conversion to sediment yield does not account for any deposition within the channels and therefore underestimates the spatially explicit sediment supply rates, but likely corresponds to the correct order of magnitude within the context of sediment budgeting technology (e.g., Reid and Dunne 1996)."

We agree with the referee that patterns of erosion often have a power or exponential distribution, however, because we do have the data to inform that more complex curve, we simply assume a linear relationship between GEP and sediment yield. In the text here we add: "To limit assumptions, we linearly scaled the independently estimated sediment yield rate to GEP values."

section 3.5: In lines 19 and 23 (also Fig. 4, 5 and 6), I think that what you're computing is the specific sediment yield, because you divide the sediment yield by the upstream drainage area. Furthermore, please be explicit here if you divide by the total upstream drainage area or only by the 'buffered' part of it. For the channel segments (Fig. 6), would it make sense to report and discuss sediment yield instead of specific sediment yield ?

We have added text that this is specific sediment yield (sediment yield at a given location within the watershed), and that the drainage area used is for the buffered portion of the stream network. We also added text to the figure captions that this describes specific sediment yield.

section 3.6: Similarly to the vegetation height index, you could report here the range of the storage potential index within the study area

**We have added text giving an example of the storage potential index calculation, and maximum value, similar to the vegetation height index.**

section 4.1: p7l10 suggests that you selected c. 50 percent of the catchment area for visual qualitative validation. How did you infer 'much more stable banks' from satellite imagery, except by the presence of vegetation ? Strictly speaking: As vegetation is detected by LiDAR data analysis and included in your index, you simply observe the effect of the index computation (what would be a verification of your approach) rather than validating the GEP index by observing eroded vs. stable banks. You could shortly discuss here a validation in similar, perhaps neighbouring catchments with available sediment measurements to check the transferability of your index at least for comparable catchments.

Validating GEP with data from multiple sediment gages is an excellent idea, however, we do not vet have the long term sediment data to make such an analysis in this basin or adjacent areas. We have revised this section to clarify: "Viewing the GEP layer draped over high resolution aerial imagery along roughly 50% of the channel network, we consistently observed steep eroding banks (bare of vegetation) in areas with high GEP values throughout the watershed. Similar observations were confirmed during 2 days of field work throughout the watershed, and in addition we consistently observed much more stable banks with vegetation in areas with lower GEP values (Table 1, Fig. 3, also see extensive photo documentation in Bigelow et al. 2012a). In the field, higher GEP values generally corresponded to steeper more convergent terrain, while lower GEP values corresponded to flatter and divergent terrain. These observations qualitatively indicate that GEP provides reasonable estimates of relative erosion within a watershed. To quantitatively access the accuracy of GEP requires multiple long-term sediment gages within a watershed, which is not vet possible in this watershed. However, when comparing erosion predictions to field inventories in the Oregon Coast Range, the index performed better than hillslope gradient alone or other available erosion models (Miller and Burnett 2007). As mentioned previously, GEP is a relative measure of erosion within a basin, which alone is highly useful to practitioners, and the conversion to sediment yield is intended to give more meaningful values that likely corresponds to the correct order of magnitude within the context of sediment budgeting technology (e.g., Reid and Dunne 1996)."

section 4.2: A question to discuss: A patch of high GEP-pixels would only be contributing if there is a continous sediment pathway towards the channel. Can you exclude that there are intermediary flat areas or other buffers that decouple the sediment source (which your index may correctly identify) and the channel? How do you deal with the channels that you know are aggrading ? I like the discussion of average yield vs episodic character of erosion.

We assume all sediment is delivered to the channel since our analysis is focused on the channel banks. We have added the following text to section 3.2 on GEP methods to clarify: "Because we confine the analysis to the buffered incised channel network, we assume that all sediment eroded from the inner gorges, arroyos, and gullies is delivered directly to the channel network, as confirmed during two days of field observations across the watershed. When practitioners want to include other forms erosion further from the channel, the proportion of sediment delivered to the channel should be estimated based on other topographic attributes (e.g. slope and confinement, Miller and Burnett 2007), which is possible within the NetMap terrain mapping platform (Benda et al. 2007) and other similar approaches."

Referee #1 had a similar question regarding aggrading channels, which we do not account for, and therefore the sediment yield conversion is over estimated. In section 3.4 we have added the following text to clarify, "The conversion to sediment yield does not account for any deposition within the channel and therefore underestimates the spatially explicit sediment supply rates, but likely corresponds to the correct order of magnitude within the context of sediment budgeting technology (e.g., Reid and Dunne 1996)."

section 4.3: Check if 'summed and area weighted' sediment yields could not be better termed 'specific sediment yields'. In 18, I suggest you replace 'total load' by 'total sediment yield at the outlet'.

**We have made these recommended changes to the text.**

section 4.4: This section is particularly important for practicioners who want to implement the approach. In the first subsection, please add a sentence on whether and how also the 'raw' i.e. pixel-scale index could be useful once a river reach or subwatershed has been identified as priority. The second subsection is too short I think, because the results of the storage index are not presented in any detail; you could, for example, highlight one or two parts of the catchment where the storage potential is particularly high or low, and explain why. Moreover, the question pops up how the floodplains that should be reconnected are disconnected at the time being; you could include this information in the study area section, for example, or where you explain the storage potential index. What measures would have to be taken to reconnect floodplains to (heavily incised !) channels ?

We have added a reference to the pixel scale index as recommended. We've also added a figure example of sediment storage potential and how the channel could be reconnected to the floodplain here.

chapter 5: Personally, I would prefer if the two proposals for potential improvements were part of the discussion section, with a (slightly longer) proper conclusion

Good idea, we have separated out and expanded a traditional conclusion section.

chapter 5. Regarding the second improvement: Can one really think of a (even average) ratio bedload vs suspended load, given different forcing magnitudes, and also given the episodic character of events in the study area that has already been mentioned ? Does the ratio of fine and coarse sediment within the sediment sources equal the ratio of fluvial sediment in transport or at the outlet ? References to that end ?

To avoid getting into the specifics and difficulties on how to account for the bedload component of sediment supply, we have reduced the detail here and simplified this section recommending that some thought and attention should be give to softer formations that may be producing more bedload relative to harder formations. It's true that in century scale events, the valley bottom channels will fill with sediment, but we were considering the bedload component from chronic scale erosion; we have also added this note to the text.

p9 l34f: The reviewer explicitly joins the authors in this statement of gratitude ;-)

;-)

**Technical Corrections:**

We have made all these technical corrections below.

- p1 125: 'increase transport capacity relative to...' instead of 'increase transport relative to...'
- p2 l2: delete 'could be applied'
- p2 114: (GEP=Generic Erosion Potential)
- p2 116: downstream
- p3 15: either 'sediment...is now supplied' or 'sediments...are now supplied'
- p5 14: Split the sentence here: 'or infer generalized relationships. For example, Pelletier...'
- p6 119: write 'outlet' rather than 'bottom'

• p6 120: might be picky, and might be due to me not being a native speaker: Sediment is not eroding, it is being transported or transferred to a channel reach, after having been eroded or mobilised.

**Referee 3: anonymous**

**Overview**:**

The authors use geospatial analysis to identify "hotspots" of erosion and sediment delivery in an incised channel network within a tributary to the San Francisco Bay in California, USA. Channel adjacent sediment sources are identified and quantified based on a combination of DEM-derived slope gradient and morphology, as well as the density and size of riparian vegetation. Rates of sediment delivery are constrained by measured sediment yield data for the watershed. Rates are summarized at the 30-m reach scale and the subbasin scale to demonstrate spatially explicit sediment delivery within the larger watershed.

**General Comments:**

I agree with anonymous referee #2 that this is a relevant and interesting study, which can provide restoration practitioners with some relatively simple tools to help identify and prioritize restoration activities in a cost-effective manner. I also agree that more information is needed on the assumptions and methods used in the analysis. Overall, I recommend this paper to be approved if additional detail is added to the manuscript.

We thank the referee for the constructive review comments. We have addressed the specific comments below and incorporated the changes into the manuscript. We have added more details on the assumptions and methods, see the response to comments by referee #2 as needed.

**Specify and Discuss Hydraulic Geometry Relationships for Modeled Watershed -** The authors cite Dunne and Leopold (1978) for hydraulic geometry relationships related to predicting bankfull width and depth. I suggest that the authors state both equations in the manuscript. Also, given that Dunne and Leopold (1978) only published the regression lines and not the data used to derive the relationships, I wonder how accurate these regional regressions are for predicting bankfull width and depth for the modeled watersheds. This is important, since the channel buffer width (i.e., source areas) is dependent on the bankfull channel width. As such, the authors should briefly describe whether the hydraulic geometry relationships from Dunne and Leopold (1978) predicted reasonable values for bankfull width and bankfull depth for the modeled watershed(s).

We have added the equations as suggested. We have also added text here, clarifying: "By overlaying several buffer sizes (2, 4, 6 times bankfull width) on the Light Detection and Ranging (LiDAR) DEM and checking them in the field, we found the 6 times bankfull width buffer was best suited to maximize the inclusion of both eroding banks of incised channels and hillslope areas that contribute sediment annually (e.g. toes of earthflows), but exclude areas that are not directly connected to channels."

**More Detail Needed on the Channel Buffering Technique -** The manuscript cites Perroy et al. (2010) as the rationale for using the buffer of 6 times the bankfull width. The Perroy paper mentions that their buffering technique scaled with stream order, but did not discuss a specific factor of 6. More information is needed on how the authors came up with the value of 6 times the bankfull width to determine their channel buffer width. Ideally, the width of the buffer should relate to the height of the bank and the geotechnical properties of the bank material, as

these properties will largely dictate the extent of the adjacent hillslope subject to failure.

We added text here, clarifying, "By overlaying several buffer sizes (2, 4, 6 times bankfull width) on the Light Detection and Ranging (LiDAR) DEM and checking them in the field, we found the 6 times bankfull width buffer was best suited to maximize the inclusion of both eroding banks of incised channels and hillslope areas that contribute sediment annually (e.g. toes of earthflows), but exclude areas that are not directly connected to channels."

**More Detail Needed on How Erosion Reduction Equation was Derived** – There needs to be more detail on the how the curve (Figure 2) and the equation (Equation 2) were created.

We have added text here clarifying, "Based on one day of field observations across the watershed where the vegetation height and associated erosion reduction (root spread) were visually estimated, we plotted and derived the best fit equation (Eq. 3, Fig. 2) that governs how erosion potential is reduced by vegetation height and use that to scale the GEP index as described below....."

**Erosion Source Areas and Proximity to Channel Knickpoints** – This might be beyond the scope of this paper, but it would be interesting to see if the erosion patterns were related to the location of discontinuities in the channel profile. Some recent literature has suggested that inner gorge failure (i.e., bank failure) happens pervasively downstream of channel knickpoints/knickzones (Bennett et al., 2016). This could give restoration practitioners even more insight on where to concentrate their restoration activities.

This is an excellent idea for analysis. We did look at a few tributary long profiles and there are some knickpoints, possibly controlled by earthflows pinching the tributary valley floor. However, as you mentioned, this analysis is beyond the scope of our study, particularly considering the watershed is 573 km2. Also there are multiple forcing mechanisms and periods for historic channel incision, including tectonic uplift, channelization of the valley floors, and changes in vegetation and runoff. In our initial study for the Zone 7 Water Agency, we advised further study to understand the controls and cycles of incision in the watershed, because as you indicate, it is fundamental to guiding restoration activities. We have added this to the discussion in the manuscript. Thank you for the reference.

**References:**

Bennett, G.L., S.R. Miller, J.J. Roering, and D.A. Schmidt. 2016. Landslides, threshold slopes, and the survival of relict terrain in the wake of the Mendocino Triple Junction. Geology. doi:10.1130/G37530.1.

---

## Editor Comment (EC1) · G. Sofia (Editor) · 16 May 2016

Dear Authors, I have now examined the discussion on your paper entitled "Delineating incised stream sediment sources within a San Francisco Bay tributary basin". I agree with the suggestions given by the reviewers, especially on the part about the assessment of erosion through DEMs of Differences (DOD) (thus the literature suggested, including also (Lane et al. 2003) in addition to (Wheaton et al. 2009; Wheaton et al. 2010). I also agree with the fact that the method part should be clarified more. As raised also during the review, i have one more question about the channel width size, I understand channel width is calculated from a regional regression based on drainage area (please cite such equation), however what is the goodness of the fitting of such equation to actual field-surveyed channel size in your study area? could you provide this information? This is important, since the proposed channel buffer width is depen-

dent on the bankfull channel width. The potential decoupling of the sediment respect to the network is also an interesting question raised during the review. I suggest the authors to consider for example (Cavalli et al. 2013, further investigated in Trevisani and Cavalli 2016) as another example to account for sediment connectivity. As one of the reviewers underlined, also the description of the channel network delineation needs clarification. You provided a series of replies which give a first overview of the steps you are going to take for the review. If you are willing to pursue these revisions, I will be pleased to reconsider your submission, with the help of the same reviewers who examined the present work. In submitting your revised version, please provide a detailed list of the changes made to the text, and a detailed list of your responses to each reviewer's comment. Please note that this editorial decision does not guarantee that your paper will be accepted for final publication in ESurf. A decision will be made only when the revised version will be available, and will be evaluated. Best regards Giulia Sofia

Cavalli M, Trevisani S, Comiti F, Marchi L (2013) Geomorphometric assessment of spatial sediment connectivity in small Alpine catchments. Geomorphology 188:31–41. doi: 10.1016/j.geomorph.2012.05.007 Lane SN, Westaway RM, Hicks DM (2003) Estimation of erosion and deposition volumes in a large, gravel-bed, braided river using synoptic remote sensing. Earth Surf Process Landforms 28:249–271. doi: 10.1002/esp.483 Trevisani S, Cavalli M (2016) Topography-based flow-directional roughness: potential and challenges. Earth Surf Dyn 4:343–358. doi: 10.5194/esurf-4-343-2016 Wheaton JM, Brasington J, Darby SE, Sear DA (2009) Accounting for uncertainty in DEMs from repeat topographic surveys: improved sediment budgets. Earth Surf Process Landforms n/a–n/a. doi: 10.1002/esp.1886 Wheaton JM, Brasington J, Darby SE, Sear DA (2010) Accounting for uncertainty in DEMs from repeat topographic surveys: Improved sediment budgets. Earth Surf Process Landforms 35:136–156. doi: 10.1002/esp.1886

---

## Author Response (AR1)

**Authors' response to referee comments on "Delineating incised stream sediment sources within a San Francisco Bay tributary basin"**

We provided an initial response to referee comments, these responses below are a little more detailed and reflect what is in the revised manuscript and also include responses to comments from the associate editor.

Responses are *italicized*

**Referee 1: Lorenzo Marchi**

**General comments**:
The authors present a set of GIS tools for the analysis of erosion and sediment yield at river basin scale and propose an application to a 573 km$^2$ watershed in California. The tools developed are easily applicable and could help watershed managers in objectively prioritizing sediment sources and recognizing areas for sediment storage. I wish to recommend some refinements in the introduction and in the description of methods

*We thank the referee for the constructive comments and suggested refinements, we have incorporated them all into the paper.*

**Specific comments**:
Section 1: Introduction and Objective

Sediment trapping by dams and gravel mining could be mentioned amongst the processes that cause channel incision (Kondolf, 1997; Surian and Rinaldi, 2003).

*We added this suggestion and references to the introduction.*

More papers (e.g., Brasington et al., 2000, Fuller et al., 2003; Picco et al., 2013) on the assessment of channel erosion and aggradation by means of DEM differencing could be cited, including the work by Wheaton et al. (2010), which provided an important advance in uncertainty assessment.

*We added these references as suggested to the introduction.*

Section 3: Methods

Line 23 of page 4 enounces that Eq. 1 employs slope gradient and local topographic convergence: the variable S, which apparently corresponds to slope, could be clearly defined.

*We have added text here to clarify, "S is slope gradient (m/m)…"*

More details could be provided on the variation of the parameter b with topographic convergence.

*We have added more details on b here: "b provides an estimate of the total contour length crossed by flow exiting a pixel. Local ccontributing area $a_L$ is calculated using the D-infinity flow direction algorithm (Tarboton 1997), which allows downslope dispersion. The flow direction for each pixel is calculated using each of eight triangular facets defined by a DEM grid point and the eight adjacent points, where values range from 0 - 8. For each facet having flow out of the pixel, we use the projection of flow direction on the exterior facet edge as a measure of contour length crossed by flow exiting the pixel from that facet. These projection lengths are summed over all edges with outgoing flow to provide an estimate of contour length b for flow exiting the pixel, where values range from 0 - 4. Contour length for planar flow is one pixel width, for divergent flow it is greater than one pixel width, and for convergent flow it is less than one pixel width. The ratio $a_L/b$ therefore incorporates effects of topographic convergence (Miller and Burnett 2007)."*

Page 5, lines 24-25: Eq. 2 simulates that grasses are less effective…" This is a rather strong simplification of the complex interactions between vegetation cover and erosion. What evidences support this ranking in the effectiveness of vegetation types in the studied area? The authors could consider the relations of their approach with methods commonly used to evaluate the influence of vegetation on erosion, such as USLE or RUSLE C-factor.

*We have removed the line in question, and explicitly mention the RUSLE C-factor here in comparison to our approach as suggested: "For example, the Universal Soil Loss Equation (USLE) (Wischmeier and Smith 1978), the revised USLE (RUSLE) (Renard et al. 1997), and similar approaches (e.g., Booth et al. 2014) simply use a generic reduction (C-factor) based on classes of vegetation cover (e.g., crop type, forest, scrub, etc.) to adjust erosion predictions over vast areas regardless of the individual size of vegetation within the categories."*
*And later in this section it's mentioned again: "Where as an entire class of vegetation covering square kilometers would get assigned a single erosion reduction factor (C factor) using USLE and RUSLE and similar approaches, here each 4 m$^2$ pixel gets assigned a reduction factor based on vegetation height as a proxy for root spread."*

Equation 3: it can be noted that the conversion factor is equivalent sediment delivery ratio (Walling, 1983), although GEP is an index of erosion, not a quantitative measure of gross erosion.

*We assume that all sediment within the buffered incised network is delivered to the channel, so we don't view the conversion factor as equivalent to a sediment delivery ratio. When considering sediment sources further from the channel, some estimate of the sediment delivery should be included. This clarification has been added to the revised manuscript: "Because the analysis was confined to the buffered incised channel network, we assume that all sediment eroded from the inner gorges, arroyos, and gullies is delivered directly to the channel network, as confirmed during two days of field observations across the watershed. When practitioners want to include other forms of erosion further from the channel, the proportion of sediment delivered to the channel should be estimated based on different topographic attributes (e.g., Mitasova et al. 1996, Miller and Burnett 2007, Cavalli et al. 2013), which can be automated within the NetMap terrain mapping platform (Benda et al. 2007) and other similar approaches."*

Sediment yield is a measure of sediment output from a watershed: it incorporates erosion and within watershed redeposition of eroded material. It is a measure lumped at watershed scale, and is dependent on watershed size. Transposing to other sites within the catchment the value of sediment yield computed at the catchment outlet, or ascribing it to land portions, as it has been done in this paper, is thus questionable. Such a transposition could be legitimate provided that the variable is considered an index and not a direct quantification of local sediment yield. Sediment yields reported for different sectors of the studied watersheds (e.g., Figs. 4 and 5) should be presented as values of an index that enables framing the local susceptibility to erosion in the context of sediment yield assessed at the river basin scale.

*We have added to the description of the conversion factor that it does not account for any deposition within the channel, and therefore underestimates sediment supply, but likely corresponds to the correct order of magnitude in the context of sediment budgeting technology (Reid and Dunne 1996).*

Section 3.6
The small DEM cell size requires to provide some technical details on the assessment of drainage area. Channel width in most cross sections is presumably much greater than DEM cell size (2 m). If applied to single DEM cells, the algorithms for upstream watershed extraction and drainage area assessment would fail to identify the whole upstream area. Some information on how this issue was managed could be provided.

*The sediment storage potential index is not calculated for single grid cells, rather it is calculated for a stream reach, that ranges in length from 2 – 80 m (ave 30 m), and contains the attributes for the entire upstream area draining to that reach, the reach gradient, the valley width at that reach location, and the estimated channel width from regional regressions for that drainage area in order to make the calculation for the sediment storage potential index for a give stream reach. We have clarified this in the text adding, "..we developed a relative sediment storage potential index (Eq. 5) that is calculated for each stream reach…"*

References

Brasington, J., Rumsby, B.T., McVey, R.A., 2000. Monitoring and modelling morphological change in a braided gravel-bed river using high resolution GPS-based survey. Earth Surface Processes and Landforms 25, 973‖990.

Fuller, I.C., Large, A.R.G., Milan, D.J., 2003. Quantifying channel development and sediment transfer following chute cutoff in a wandering gravel-bed river, Geomorphology, 54, 307-323.

Kondolf, G.M., 1997. Hungry Water: Effects of Dams and Gravel Mining on River Channels. Environmental Management, 21, 533‖551.

Picco, L., Mao, L., Cavalli, M., Buzzi, E., Rainato, R., Lenzi, M.A., 2013. Evaluating short-term morphological changes in a gravel-bed braided river using terrestrial laser scanner. Geomorphology, 201, 323-334, DOI: 10.1016/j.geomorph.2013.07.007

Nicola Surian, Massimo Rinaldi, 2003. Morphological response to river engineering and management in alluvial channels in Italy, Geomorphology, 50, 307-326.

Walling, D.E., 1983. The sediment delivery problem. Journal of Hydrology, 65, 209-237.

Wheaton, J.M., Brasington, J., Darby, S.E., Sear, D.A., 2010. Accounting for uncertainty in DEMs from repeat topographic surveys: improved sediment budgets. Earth Surf. Process. Landf., 35, 136‒156, doi:10.1002/es

**Referee 2: anonymous**

**General Comments:**
This is an original and interesting study. The well-structured and well-written paper describes an empirical approach of assessing (potential) sediment yield using topographic and vegetation information from airborne LiDAR data. Slope, convergence and the local (!) contributing area are combined for each raster cell into a 'generic erosion potential' index. The analysis is restricted to a buffer zone around the channel network within a c. 500 sqkm catchment where incision and bank failure and erosion present issues for sediment management. Measured mean sediment yield is used to 'translate' the GEP index into estimated sediment yield that is reduced based on an empirical relationship of vegetation coverage and yield reduction; vegetation coverage is assessed using first vs last pulse returns of airborne LiDAR data. The authors use the pixel-based data for upscaling to various scales, e.g. channel reaches and subcatchments. My biggest concern is the fact that the contributing area beyond the 3x3 neighbourhood of each raster cell, and potential decoupling of sediment sources and the channel network (and also within or along the channel network) are apparently not accounted for. A more thorough and reproducible description of some of the methods used would help resolve possible misunderstandings on my part. In all, I think that moderate revisions will make this contribution an interesting paper, not only for the scientific community but especially for practicioners.

*We wish to thank the referee for their particularly thoughtful and constructive review and comments. Concerns with the methods are due primarily to our lack of detail that has now been added, and is also addressed in the referee's specific comments later. We have incorporated most of the suggestions and they have greatly improved the paper. We address the referee's biggest concerns here too, in three parts:*

*1. To clarify that generic erosion potential (GEP) is not used to characterize active incision/downcutting of channels (which would require a different method including the entire upslope drainage area, e.g. stream power), we have added the following text in the GEP methods section: "We specifically use GEP as a relative index of erosion potential for shallow failures along steep banks of incised channels in the Arroyo Mocho study area. GEP is not used to estimate erosion potential for downcutting the channel bed or bank erosion of outside river bends. As mentioned earlier, the incised channels in the Arroyo Mocho watershed do not appear to be actively downcutting, rather sediment is now supplied to most channels by shallow slides and failures of oversteepened banks (channel widening), a common stage in the evolution of incised channels following downcutting (Schumm et al. 1984)."*

*2. To address the referee's concern of using local contributing area for erosion potential, we have added text to the GEP methods explaining: "Local contributing area is used for erosion potential rather than the entire upslope drainage area because (1) conceptually, pore pressures measured during storms in shallow soils correlate poorly with topography (e.g., Dhakal and Sullivan 2014), but convergent areas tend to exhibit persistently high water content, deeper soils, and are highly responsive to rainfall events, and (2) empirically, local contributing area has been shown to better predict shallow failures than total contributing area in several studies (Miller 2004, Miller and Burnett 2007). While upslope drainage area is often used as a surrogate for subsurface flow, most locations on the hillslope receive contributions from a small*

*proportion of the upslope contributing area due to the low velocity of subsurface flow (Barling et al., 1994, Beven and Freer 2001, Borga et al. 2002)." While it seems reasonable that the degree of antecedent soil moisture may correlate with total contributing area (e.g., Hotta et al., 2010), the time scales for transmission of pressure changes in response to rainfall can span from minutes to centuries (Iverson 2000), making it difficult to discern how much of the contributing area influences pore-pressure response at any particular point during any particular storm. Intense rainfall tends to be of short duration, and shallow slides tend to be triggered by intense rainfall (with high antecedent conditions). Also, if you apply the kinematic wave approach used for Shalstab-like models, and apply a Darcy velocity for saturated flow, the upslope length of contributing area varies with storm duration.*

*3. To address the referee's concern that sediment sources decoupled from the channel may not be accounted for, we have added text to the GEP methods section clarifying: "Because the analysis was confined to the buffered incised channel network, we assume that all sediment eroded from the inner gorges, arroyos, and gullies is delivered directly to the channel network, as confirmed during two days of field observations across the watershed. When practitioners want to include other forms of erosion further from the channel, the proportion of sediment delivered to the channel should be estimated based on different topographic attributes (e.g., Mitasova et al. 1996, Miller and Burnett 2007, Cavalli et al. 2013), which can be automated within the NetMap terrain mapping platform (Benda et al. 2007) and other similar approaches." Also, we have added text explicitly recognizing that channel storage/aggradation is not accounted for: "The conversion to sediment yield does not account for any deposition within the channels and therefore underestimates the spatially explicit sediment supply rates, but likely corresponds to the correct order of magnitude within the context of sediment budgeting technology (e.g., Reid and Dunne 1996)." New references cited above:*

*Barling, D. B., Moore, I. D., and Grayson, R. B.: A quasi-dynamic wetness index for characterising the spatial distribution of zones of surface saturation and soil water content. Water Resour. Res. 30, 1029–1044, 1994.*

*Beven, K., and Freer, J.: A dynamic TOPMODEL. Hydrol. Process. 15, 1993–2011, 2001.*

*Borga, M., Dalla Fontana, G., and Cazorzi, F.: Analysis of topographic and climatic control on rainfall-triggered shallow landsliding using a quasi-dynamic wetness index, Journal of Hydrology, 268, 56-71, 2002.*

*Dhakal, A. S., and Sullivan, K.: Shallow groundwater response to rainfall on a forested headwater catchment in northern coastal California: implications of topography, rainfall, and throughfall intensities on peak pressure head generation, Hydrological Processes, 28, 446-463, 2014.*

*Hotta, N., Tanaka, N., Sawano, S., Kuraji, K., Shiraki, K., & Suzuki, M. (2010). Changes in groundwater level dynamics after low-impact forest harvesting in steep, small watersheds. Journal of hydrology, 385(1), 120-131.*

*Iverson, Richard M. "Landslide triggering by rain infiltration." Water resources research 36.7 (2000): 1897-1910.*

*Miller, D. J.: Landslide Hazards in the Stillaguamish basin: a new set of GIS tools, prepared for the Stillaguamish Tribe of Indians Natural Resource Department, 48 pp., 2004.*

**Specific Comments:**

p1 l23: common erosional cycle suggest that this is a totally natural, authigenic phenomenon; however, as you elaborate in the following lines, this kind of erosional phenomenon represents the response to environmental or anthropogenic changes. I'd like to suggest you emphasize more the 'reaction to change' aspect;if you insist on the 'common erosional cycle', please give a reference

*Good point. We have clarified that channel incision is a common erosional response to natural or anthropogenic forcing and added more references to natural causes of incision. This section now reads: "Channel incision is a common erosional response to natural or anthropogenic forcing that poses challenges to watershed management across the globe (Schumm et al. 1984, Schumm 1999, 2007). Incised channels (inner gorges, arroyos, gullies, ravines, etc.) are often created by headward incision of the channel network in response to local or regional base-level lowering (Begin et al. 1981, Schumm 1993), tectonic uplift (Burnett and Schumm 1983), or disturbances that change the relative balance between sediment transport and supply (Schumm et al. 1984, Schumm 1999). For example, incision caused by humans can include urbanization that greatly increases runoff and sediment transport (Booth 1991), or dams and gravel mining that reduce sediment supply (Kondolf 1997, Surian and Rinaldi 2003). Examples of natural causes of incision include change to a wetter climate that increases runoff (Balling and Wells 1990) and catastrophic events that increase sediment supply from volcanic eruptions (Simon 1992), extreme precipitation (Miller and Benda 2000), or following wildfire (Benda et al. 1998)."*

p2 l14: with 'shallow failures' you mean bank failures ?

*We have clarified here that we mean shallow failures from planar and convergent slopes. These are inner gorge type failures from over steepened banks, not bank erosion (e.g. outside river bends).*

p2 l16: The virtual watershed system or software is only named and explained in the methods section (p3 l 36); here, the reader might wonder if it is a purely virtual experiment , not a study using data from a real landscape. You also mention a 'synthetic river network' the reader might possibly misinterpret along the same lines. I suggest you shift the short explanation of the virtual watershed system from the methods section to the introduction.

*We removed the mention of virtual watershed and synthetic here to avoid confusion and clarify the NetMap terrain mapping platform was used to generate the various GIS layers.*

p3 l8f: if you have aggrading channel reaches in your area, then the approach of dividing the

outlet sediment yield by the size of the incised area+buffer is probably affected by this: The sum of sediment delivery from the channel banks could be larger than the measured sediment yield due to intermediate sedimentation. I feel you should discuss this where you explain the transfer from the GEP to sediment delivery

*Referee #1 had a similar concern, we have clarified in con Conversion to Sediment Yield, that "The conversion to sediment yield does not account for any deposition within the channel and therefore underestimates the spatially explicit sediment yield rates, but likely corresponds to the correct order of magnitude within the context of sediment budgeting technology (Reid and Dunne 1996)."*

p3 l28: A figure with one or more 'typical' cross-sections through the valley, maybe one in the lower and one in the upper reaches, could help visualise and justify your statement that the valley floors and hillslopes are not eroded, and the choice of your buffer width. Moreover, I could only guess that the 6times bankful width refers to the total buffer width and not its radius to both sides.)

*We have added a figure (Fig 3) with cross sections as suggested and clarified that the buffer is 6 times bankfull width is the total buffer width.*

section 3.1: the description of how you produced the channel network is too vague, it only refers to the literature; please add at least a brief description of the(major) steps involved.

*We have included more detail on the major steps used to derive the channel network and how the channel segments were delineated. This section now reads as follows: " An attributed digital stream network was derived from the DEM using a series of algorithms described by Miller (2002). The following major steps were followed in the construction of the DEM and extraction of the channel network:*

1. *DEM. The 2 m DEM was compiled and resampled from a 0.3 m DEM for the majority of the watershed within the dominant jurisdictional boundary (Alameda County) and a 3 m DEM for small portions of the watershed that lie in other counties, all derived from LiDAR data collected by the U.S. Geological Survey. The objective here was to maintain information from the most precise and accurate data in creation of a single, contiguous DEM, while also avoiding creation of breaks in elevation or derivatives of elevation (gradient, curvature) at seams between the different data sources.*

2. *Topographic attributes. Topographic attributes for network extraction were based on unaltered elevation data, prior to drainage enforcement and hydrologic conditioning. The attributes calculated are surface gradient, plan curvature, and the contour length crossed by flow out of a DEM cell, which is used to calculate local contributing area.*

3. *Channel Initiation. Channel initiation points were defined using slope-dependent drainage area threshold criteria (e.g. Montgomery and Dietrich 1992, Dietrich et al. 1993) based on flow accumulation and surface slope, plan (contour) curvature, and flow length over which these criteria are met (Miller 2002, Miller et al. 2015).*

4. *Hydrologically conditioned DEM. A hydrologically conditioned DEM was created where the flow paths out of all closed depressions are identified using a combination of depression filling (Jenson and Domingue, 1988) and carving (Soille et al., 2003).*

5. *Flow and channels. Flow accumulation was calculated to identify all channel initiation points, and trace all channels. The D-infinity algorithm (Tarboton, 1997) was applied to calculate flow accumulation values down to channel initiation points. Channels were then traced downstream from these points using D-8 flow directions (Jenson and Domingue, 1988) to preclude dispersion of channelized flow. D-8 flow directions are chosen using a combination of steepest descent and largest plan curvature (Clarke et al., 2008). Using algorithms by Miller (2002), the channel network was divided into homogeneous reaches with end-point positions selected that minimize variance of channel gradient, valley width, and drainage area within a reach, while keeping mean reach lengths no more than 80 m (ave 30 m); each reach was attributed with various topographic measures, including channel slope, valley width, drainage area, etc. (Miller 2002).*

6. *Refining Network. Channel traces were smoothed to provide better-placed channel centerlines and more accurate estimates of channel length and gradient. The delineated channel network was validated using a combination of local knowledge, field observations, and high-resolution optical imagery. "*

E.g., how did you identify the channel initiation points for the 'synthetic' network ?

*We have added text to clarify, "Channel initiation points were defined using slope-dependent drainage area threshold criteria (e.g. Montgomery and Dietrich 1992, Dietrich et al. 1993) based on flow accumulation and surface slope, plan (contour) curvature, and flow length over which these criteria are met (Miller 2002, Miller et al. 2015)."*

Another weakness here is that the reader does not know how you delineated the segments: manually ? automatically ? On what basis or using which criteria ?

*We have added text to clarify, "Using algorithms by Miller (2002), the channel network was divided into homogeneous reaches with end-point positions selected that minimize variance of channel gradient, valley width, and drainage area within a reach, while keeping mean reach lengths no more than 80 m (ave 30 m); each reach was attributed with various topographic measures, including channel slope, valley width, drainage area, etc. (Miller 2002)."*

section 3.2: l27 ff lack the description of S and its unit (percent ? degree ?).

*We have added text that "S is slope gradient (m/m)"*

You only give a reference for the derivation of b, I feel that you need to describe the major steps at least. I cannot figure out how GEP could be constrained to values between 0 and 1 as long as I do not know the range of S and b. For example, if the slope on a raster cell is steep, and if it has, say, 5 contributing neighbours, the product of S and aL is supposed to be quite large; if you divide it by b which is only said to be 'around 1', GEP will be larger than 0, right ?

*We have added more detail on local contributing $a_L$ and b that "Local contributing area $a_L$ is calculated using the D-infinity flow direction algorithm (Tarboton 1997), which allows downslope dispersion. The flow direction for each pixel is calculated using each of eight triangular facets defined by a DEM grid point and the eight adjacent points, where values range from 0 - 8. For each facet having flow out of the pixel, we use the projection of flow direction on the exterior facet edge as a measure of contour length crossed by flow exiting the pixel from that facet. The projection lengths are summed over all edges with outgoing flow to provide an estimate of contour length b for flow exiting the pixel, where values range from 0 - 4. Contour length for planar flow is one pixel width, for divergent flow it is greater than one pixel width, and for convergent flow it is less than one pixel width……….. Thus GEP is a dimensionless relative index of erosion potential with most values within a watershed ranging from 0 – 1,…"*

Moreover, the contributing area of a raster cell can be MUCH larger than just the up to 8 surrounding pixels, what about the mouth of a small tributary rill cutting the riverbank ? Above, I understood that you're linking the contributing areas to the channel network using flow accumulation (and direction) within the buffer zone; and I do not think you can exclude the contributing area of a cell when assessing erosion potential (c.f. stream power index)...

*We have added text that GEP is not used for characterizing downcutting or erosion of outside river bends, for which you would need to include a model that accounts for stream power (the entire upstream drainage area). We only use GEP to characterize shallow failures from the steep eroding banks of incised channels. We use local contributing area, because it has been shown to predict shallow failures better. We have accordingly added text to clarify: "We specifically use GEP as a relative index of erosion potential for shallow failures along steep banks of incised channels in the Arroyo Mocho study area. GEP is not used to estimate erosion potential for downcutting the channel bed or bank erosion of outside river bends. As mentioned earlier, the incised channels in the Arroyo Mocho watershed do not appear to be actively downcutting, rather sediment is now supplied to most channels by shallow slides and failures of oversteepened banks (channel widening), a common stage in the evolution of incised channels following downcutting (Schumm et al. 1984)." and "Local contributing area is used for erosion potential rather than the entire upslope drainage area because (1) conceptually, pore pressures measured during storms in shallow soils correlate poorly with topography (e.g., Dhakal and Sullivan 2014), but convergent areas tend to exhibit persistently high water content, deeper soils, and are highly responsive to rainfall events, and (2) empirically, local contributing area has been shown to better predict shallow failures than total contributing area in several studies (Miller 2004, Miller and Burnett 2007). While upslope drainage area is often used as a surrogate for*

*subsurface flow, most locations on the hillslope receive contributions from a small proportion of the upslope contributing area due to the low velocity of subsurface flow (Barling et al., 1994, Beven and Freer 2001, Borga et al. 2002)."*

Moreover: Would you agree that a pit with 8 contributing neighbours is prone to erosion ? Well, the slope is 0 inside a pit, but depending on the slope derivation algorithm this is not always the case. It may be that I misunderstood parts of the description, but I feel it has to be more extensive in order to be reproducible; the reader shouldn't have to read Miller and Burnett (2007) first.

*In the example of a pit, the GEP walls would get a high value and the floor a low value or 0. Hopefully the details added on GEP methods now make this clear.*

section 3.3: I can understand the need to include a parameterisation for vegetation effects, and deriving this using the LiDAR first+last signal approach is fine for me. However, I think that how you arrive at the erosion reduction factor is poorly describe. You mention 'supporting literature' and 'own observations', but it is not clear how you got the percentage reductions used in Fig 2 to derive equation 2.

*We have added text to clarify: "During one day of field observations across the watershed, we observed less erosion on banks with taller trees (and larger root spread) compared to banks with smaller trees and shrubs. Here, we visually estimated the average riparian vegetation height on a given bank and the proportion of the bank eroded, the inverse of the latter gives an estimate of erosion reduction. These observations were plotted to derive the best fit equation (Eq. 3, Fig. 4) that governs how erosion potential is reduced by vegetation height…"*

In line 26 I can only guess why the vegetation height derived from first+last pulse should also represent vegetation density;

*Although we still believe a measure of vegetation density is represented by the vegetation height grid, we couldn't clarify it any more than previously written ("Because the tree canopy height of each 2 m grid cell is represented, this grid also represents the density of vegetation, another factor that can reduce erosion"), so we have removed any reference to vegetation density.*

please write a sentence to explain. Line 31: In my opinion you do not normalize the vegetation height, but you apply equation 2 to the 'elevation height' raster.

*You are correct. We have revised this to read: "To reduce GEP based on this relationship, a vegetation height grid (2 m) was created in GIS using the first (representing the tallest vegetation) and last (representing the ground surface) return LiDAR points. An erosion reduction grid was then created by multiplying the vegetation height grid by Eq. 3."*

section 3.4: As stated earlier, I feel you should be (more) explicit about some assumptions: (i) within the buffer zone, no intermediary barriers or buffers exist that decouple sediment sources from the channels (ii) there is no intermediary deposition or aggradation on the longer timescale within the whole channel network. This (at least) has to be assumed, because the measured sediment yield at the output is related to the entire size of the buffer zone. However, I

agree that such simplifications have to be implemented, especially when the aim of setting up a practicioner-friendly tool.

*We have added text to this section, clarifying: "Because the analysis was confined to the buffered incised channel network, we assume that all sediment eroded from the inner gorges, arroyos, and gullies is delivered directly to the channel network, as confirmed during two days of field observations across the watershed. When practitioners want to include other forms of erosion further from the channel, the proportion of sediment delivered to the channel should be estimated based on different topographic attributes (e.g., Mitasova et al. 1996, Miller and Burnett 2007, Cavalli et al. 2013), which can be automated within the NetMap terrain mapping platform (Benda et al. 2007) and other similar approaches."*

*We have also added text on the conversion to sediment yield, clarifying that, "The conversion to sediment yield does not account for any deposition within the channels and therefore underestimates the spatially explicit sediment supply rates, but likely corresponds to the correct order of magnitude within the context of sediment budgeting technology (e.g., Reid and Dunne 1996)."*

Moreover: Please reconsider if a linear relationship of GEP and sediment yield is appropriate. Statistical relationships contributing area and discharge or sediment yield have a sort of power law...

*We agree with the referee that patterns of erosion often have a power or exponential distribution, however, because we do not have the data to inform that more complex curve, we simply assume a linear relationship between GEP and sediment yield. In the text here we add: "To limit assumptions, we linearly scaled the independently estimated sediment yield rate to GEP values."*

section 3.5: In lines 19 and 23 (also Fig. 4, 5 and 6), I think that what you're computing is the specific sediment yield, because you divide the sediment yield by the upstream drainage area. Furthermore, please be explicit here if you divide by the total upstream drainage area or only by the 'buffered' part of it. For the channel segments (Fig. 6), would it make sense to report and discuss sediment yield instead of specific sediment yield ?

*We have added text that this is specific sediment yield (sediment yield at a given location within the watershed), and that the drainage area used is for the buffered portion of the stream network. We also added text to the figure legends and captions that this describes specific sediment yield.*

section 3.6: Similarly to the vegetation height index, you could report here the range of the storage potential index within the study area

*We have added text giving an example of the storage potential index calculation, and the range of values.*

section 4.1: p7l10 suggests that you selected c. 50 percent of the catchment area for visual qualitative validation. How did you infer 'much more stable banks'from satellite imagery, except

by the presence of vegetation ? Strictly speaking: As vegetation is detected by LiDAR data analysis and included in your index, you simply observe the effect of the index computation (what would be a verification of your approach) rather than validating the GEP index by observing eroded vs. stable banks.

*We actually used both air photos and field observations, we have added text to clarify: "Viewing the GEP layer draped over high resolution aerial imagery along roughly 50% of the channel network, we consistently observed steep eroding banks (bare of vegetation) in areas with high GEP values throughout the watershed. Similar observations were confirmed during 2 days of fieldwork, where additionally we consistently observed much more stable banks with vegetation in areas with lower GEP values (Table 1, Fig. 5, also see extensive photo documentation in Bigelow et al. 2012a). In the field, higher GEP values generally corresponded to steeper more convergent terrain, while lower GEP values corresponded to flatter and divergent terrain. These observations qualitatively indicate that GEP provides reasonable estimates of relative erosion within a watershed."*

You could shortly discuss here a validation in similar, perhaps neighbouring catchments with available sediment measurements to check the transferability of your index at least for comparable catchments.

*Validating GEP with data from multiple sediment gages is an excellent idea, however, we do not yet have the long term sediment data to make such an analysis in this basin or adjacent areas. We have revised this section to clarify: "To quantitatively access the accuracy of GEP requires multiple long-term sediment gages within a watershed, which is not yet possible in this watershed. However, when comparing erosion predictions to field inventories in the Oregon Coast Range, the index performed better than hillslope gradient alone or other available erosion models (Miller and Burnett 2007). As mentioned previously, GEP is a relative measure of erosion within a basin, which alone is highly useful to practitioners, and the conversion to sediment yield is intended to give more meaningful values that likely corresponds to the correct order of magnitude within the context of sediment budgeting technology (e.g., Reid and Dunne 1996)."*

section 4.2: A question to discuss: A patch of high GEP-pixels would only be contributing if there is a continous sediment pathway towards the channel. Can you exclude that there are intermediary flat areas or other buffers that decouple the sediment source (which your index may correctly identify) and the channel?

*We assume all sediment is delivered to the channel since our analysis is focused on the channel banks. We have added the following text to the GEP methods to clarify: "Because the analysis was confined to the buffered incised channel network, we assume that all sediment eroded from the inner gorges, arroyos, and gullies is delivered directly to the channel network, as confirmed during two days of field observations across the watershed. When practitioners want to include other forms of erosion further from the channel, the proportion of sediment delivered to the channel should be estimated based on different topographic attributes (e.g., Mitasova et al. 1996, Miller and Burnett 2007, Cavalli et al. 2013), which can be automated within the NetMap terrain mapping platform (Benda et al. 2007) and other similar approaches."*

How do you deal with the channels that you know are aggrading ? I like the discussion of average yield vs episodic character of erosion.

*Referee #1 had a similar question regarding aggrading channels, which we do not account for, and therefore the sediment yield conversion is over estimated. In the section on conversion to sediment yield we have added the following text to clarify, "The conversion to sediment yield does not account for any deposition within the channel and therefore underestimates the spatially explicit sediment supply rates, but likely corresponds to the correct order of magnitude within the context of sediment budgeting technology (e.g., Reid and Dunne 1996)." .*

section 4.3: Check if 'summed and area weighted' sediment yields could not be better termed 'specific sediment yields'. In l8, I suggest you replace 'total load' by 'total sediment yield at the outlet'.

*We have made these recommended changes to the text.*

section 4.4: This section is particularly important for practicioners who want to implement the approach. In the first subsection, please add a sentence on whether and how also the 'raw' i.e. pixel-scale index could be useful once a river reach or subwatershed has been identified as priority.

*We added a sentence here clarifying how the pixel scale maps can be used: "After a subwatershed has been prioritized for source control, the reach and pixel scale maps of erosion (Fig. 5) can be used to target specific valley segments, reaches, and banks within the subwatershed."*

The second subsection is too short I think, because the results of the storage index are not presented in any detail; you could, for example, highlight one or two parts of the catchment where the storage potential is particularly high or low, and explain why.

*We have expanded this section as suggested, with the following text: "While valley floor channels with high potential for sediment storage located throughout the watershed (Fig. 10), there are more undeveloped valley floors in the eastern and southern portions of the basin, and locations here may be more conducive to restoration projects. As an example application of the sediment storage potential index in this region, Fig 11. shows an ideal area for sediment storage in lower Cayetano Creek, where the channel could be plugged, forcing flow onto the former floodplain, or with a series of ponds and plugs, where the channel is widened (creating ponds) and the borrow material is used to plug the channel (e.g. Rosgen 1997). The sediment storage potential here is moderately high because the valley is wide and the channel slope is low. In addition to sediment supply and sediment storage potential, practitioners should also consider the causes, current evolutionary stage, and history of channel incision within the watershed when determining the location of restoration projects to promote sediment storage (Schumm 1999). The Arroyo Mocho basin has undergone several cycles of incision from tectonic uplift (Sloan 2006), channelization of valley bottoms (Williams 1912), and changes in vegetation and runoff (Rogers 1988). Still, the incision cycles, history, and causes remain poorly understood*

*and such knowledge would better target restoration locations and perhaps prevent or mitigate future cycles of incision from anthropogenic causes."*

Moreover, the question pops up how the floodplains that should be reconnected are disconnected at the time being; you could include this information in the study area section, for example, or where you explain the storage potential index.

*We have added this information to the text: "Today, most valley floor channels are physically disconnected from their floodplain by channelization (engineered channels or ditches) and channel incision that migrated upstream from that channelization (base level lowering) and other causes."*

What measures would have to be taken to reconnect floodplains to (heavily incised !) channels?

*We have added text here giving an example of how to reconnect channels to floodplains, "..Fig 11. shows an ideal area for sediment storage in lower Cayetano Creek, where the channel could be plugged and diverted onto a new channel created near grade with the former floodplain, or with a series of ponds and plugs, where the channel is widened (creating ponds) and the excavated material from the channel widening is used to plug the channel (e.g. Rosgen 1997)"*

chapter 5: Personally, I would prefer if the two proposals for potential improvements were part of the discussion section, with a (slightly longer) proper conclusion

*Good idea, we have separated out and expanded a more traditional summary and conclusion section.*

chapter 5. Regarding the second improvement: Can one really think of a (even average) ratio bedload vs suspended load, given different forcing magnitudes, and also given the episodic character of events in the study area that has already been mentioned ? Does the ratio of fine and coarse sediment within the sediment sources equal the ratio of fluvial sediment in transport or at the outlet ? References to that end ?

*To avoid getting into the specifics and difficulties on how to account for the bedload component of sediment supply, we have reduced the detail here, eliminated recommending any specific approach, but simply recommend that some thought and attention should be given to softer formations that may be producing more bedload relative to harder formations. Here the text now reads: "While approaches characterizing bedload yield variation across a watershed can range in complexity (e.g., Dietrich and Dunne 1978, Collins and Dunne 1989, Benda and Dunne 1997, Madej 1995, O'Connor et al. 2014), a simplified approach is likely appropriate considering that limitations of sediment budgeting technology typically constrain estimates to the correct order of magnitude (e.g., Reid and Dunne 1996)."*

*It's true that in century scale events, the valley bottom channels will fill with sediment, but we were considering the bedload component from chronic scale erosion; we have also added this note to the text in an earlier section.*

p9 l34f: The reviewer explicitly joins the authors in this statement of gratitude ;-)

*;-)*

**Technical Corrections:**

*We have made all these technical corrections below, except where noted.*

• p1 l25: 'increase transport capacity relative to...' instead of 'increase transport relative to...'

*This now reads*: "*or disturbances that change the relative balance between sediment transport and supply*"

• p2 l2: delete 'could be applied'

• p2 l14: (GEP=Generic Erosion Potential)

• p2 l16: downstream

• p3 l5: either 'sediment...is now supplied' or 'sediments...are now supplied'

• p5 14: Split the sentence here: 'or infer generalized relationships. For example, Pelletier...'

• p6 l19: write 'outlet' rather than 'bottom'

• p6 l20: might be picky, and might be due to me not being a native speaker: Sediment is not eroding, it is being transported or transferred to a channel reach, after having been eroded or mobilised.

*We changed this to "sediment supplied…."*

**Referee 3: anonymous**

**Overview**:
The authors use geospatial analysis to identify "hotspots" of erosion and sediment delivery in an incised channel network within a tributary to the San Francisco Bay in California, USA. Channel adjacent sediment sources are identified and quantified based on a combination of DEM-derived slope gradient and morphology, as well as the density and size of riparian vegetation. Rates of sediment delivery are constrained by measured sediment yield data for the watershed. Rates are summarized at the 30-m reach scale and the subbasin scale to demonstrate spatially explicit sediment delivery within the larger watershed.

**General Comments**:
I agree with anonymous referee #2 that this is a relevant and interesting study, which can provide restoration practitioners with some relatively simple tools to help identify and prioritize restoration activities in a cost-effective manner. I also agree that more information is needed on the assumptions and methods used in the analysis. Overall, I recommend this paper to be approved if additional detail is added to the manuscript.

*We thank the referee for the constructive review comments. We have addressed the specific comments below and incorporated the changes into the manuscript. We have added more details on the assumptions and methods, see the response to comments by referee #2 and the revised manuscript as needed.*

**Specify and Discuss Hydraulic Geometry Relationships for Modeled Watershed -** The authors cite Dunne and Leopold (1978) for hydraulic geometry relationships related to predicting bankfull width and depth. I suggest that the authors state both equations in the manuscript. Also, given that Dunne and Leopold (1978) only published the regression lines and not the data used to derive the relationships, I wonder how accurate these regional regressions are for predicting bankfull width and depth for the modeled watersheds. This is important, since the channel buffer width (i.e., source areas) is dependent on the bankfull channel width. As such, the authors should briefly describe whether the hydraulic geometry relationships from Dunne and Leopold (1978) predicted reasonable values for bankfull width and bankfull depth for the modeled watershed(s).

*We have added the equations and reference to a study downstream that evaluated the regional regression. Note that we only use the relationship as one that scales regionally with drainage area to establish a buffer surrounding the incised channels, not to define the channel edges. Alternatives of hand digitizing the incised channel forms or developing code to automate the procedure were not feasible considering the massive effort needed. We did try various factors of bankfull widths as buffers (2x, 4x, 6x) and checked them in the field and overlays of shaded relief and air photos to see which buffer fit best.*

*We have added this to the text, including a new figure (Fig. 3) that gives an example of the buffer fit: "To use a buffer width that scales regionally with drainage area, we used a GIS buffer width of 6 times the total bankfull channel width that captures the steep eroding banks of the incised channel form and areas immediately adjacent to the channel (Fig. 3). To estimate bankfull channel width, we used a San Francisco Bay Area regional regression relationship based on*

*drainage area (Dunne and Leopold 1978):*

$$bankfull\ channel\ width\ (m)\ =\ 3.3494\ (drainage\ area\ [km^2])^{0.3737} \hspace{2cm} (1)$$

*In comparison to field measurements downstream from Arroyo Mocho, Bigelow et al. 2008 found the regional regression slightly over estimated actual bankfull width. The channel width estimate used here is only to create a buffer capturing the area around incised channels, not to define channel edges. We checked the fit of several buffer sizes (2, 4, and 6 times total bankfull width) on the shaded relief DEM (Fig. 3), air photos, and in the field, and found the 6 times total bankfull width buffer was best suited to maximize the inclusion of both eroding banks of incised channels and hillslope areas that contribute sediment annually (e.g. toes of earthflows), but exclude areas that are not directly connected to channels. A small area of valley floor or hillslope of generally low erosion potential beyond the channel edge is included in this buffer. This buffering to capture the incised channel network still provides the most reasonable solution, considering the massive effort needed for alternatives to either digitize incised channels over a 573 km² watershed or develop code to automate the procedure...."*

*Similarly, with using bankfull depth in the sediment storage potential index, we are only using it to give some height above the channel that scales with drainage area regionally at which to measure valley width, and we tried various factors (1x, 2x, 3x) of bankfull depth to see which measure of valley width gave the best results. Outside of field measurements, which we do not have, we are unaware of alternatives to regional regressions to estimate channel width and depth. We added the equation to the text and a reference to a downstream study that evaluated the regional regression. Here, the text now reads: "In comparison to field measurements downstream from Arroyo Mocho, Bigelow et al. 2008 found the regional regression estimates were similar and occasionally slightly lower than actual bankfull depth and slightly underestimated bankfull width. The bankfull depth estimate is only used to provide some height above the channel that scales regionally with drainage area at which to measure valley width....We calculated the sediment storage potential index with valley widths at 1- 3 times the bankfull depth, and found valley width at 2 times the bankfull depth produced the most varied and best results in our basin."*

**More Detail Needed on the Channel Buffering Technique -** The manuscript cites Perroy et al. (2010) as the rationale for using the buffer of 6 times the bankfull width. The Perroy paper mentions that their buffering technique scaled with stream order, but did not discuss a specific factor of 6.  More information is needed on how the authors came up with the value of 6 times the bankfull width to determine their channel buffer width.  Ideally, the width of the buffer should relate to the height of the bank and the geotechnical properties of the bank material, as these properties will largely dictate the extent of the adjacent hillslope subject to failure.

*We have deleted the Perroy et al. 2010 reference as a rationale. See the response above for the buffering technique rationale.*

**More Detail Needed on How Erosion Reduction Equation was Derived –** There needs to be more detail on the how the curve (Figure 2) and the equation (Equation 2) were created.

*We have added text here clarifying, "During one day of field observations across the watershed, we observed less erosion on banks with taller trees (and larger root spread) compared to banks with smaller trees and shrubs. Here, we visually estimated the average riparian vegetation height on a given bank and the proportion of the bank eroded, the inverse of the latter gives an estimate of erosion reduction. These observations were plotted to derive the best fit equation (Eq. 3, Fig. 4) that governs how erosion potential is reduced by vegetation height…"*

**Erosion Source Areas and Proximity to Channel Knickpoints** – This might be beyond the scope of this paper, but it would be interesting to see if the erosion patterns were related to the location of discontinuities in the channel profile. Some recent literature has suggested that inner gorge failure (i.e., bank failure) happens pervasively downstream of channel knickpoints/knickzones (Bennett et al., 2016). This could give restoration practitioners even more insight on where to concentrate their restoration activities.

*Thank you for the reference, this is an excellent idea for analysis. We did look at a few tributary long profiles and there are some knickpoints, possibly controlled by earthflows pinching the tributary valley floor. However, as you mentioned, this analysis is beyond the scope of our study, particularly considering the watershed is 573 km$^2$. Also there are multiple forcing mechanisms and periods for historic channel incision, including tectonic uplift, channelization of the valley floors, and changes in vegetation and runoff. In our initial study for the Zone 7 Water Agency, we advised further study to understand the controls and cycles of incision in the watershed, because as you indicate, it is fundamental to guiding restoration activities. We have added this to the discussion in the manuscript: "In addition to sediment supply and sediment storage potential, practitioners should also consider the causes, current evolutionary stage, and history of channel incision within the watershed when determining the location of restoration projects to promote sediment storage (Schumm 1999). The Arroyo Mocho basin has undergone several cycles of incision from tectonic uplift (Sloan 2006), channelization of valley bottoms (Williams 1912), and changes in vegetation and runoff (Rogers 1988). Still, the incision cycles, history, and causes remain poorly understood and such knowledge would better target restoration locations and perhaps prevent or mitigate future cycles of incision from anthropogenic causes."*

**References:**

Bennett, G.L., S.R. Miller, J.J. Roering, and D.A. Schmidt. 2016. Landslides, threshold slopes, and the survival of relict terrain in the wake of the Mendocino Triple Junction. Geology. doi:10.1130/G37530.1.

Associate Editor: Giulia Sofia

Dear Authors, I have now examined the discussion on your paper entitled "Delineating incised stream sediment sources within a San Francisco Bay tributary basin".

*We thank the associate editor for the constructive comments provided and an earlier set of comments that improved the paper.*

I agree with the suggestions given by the reviewers, especially on the part about the assess- ment of erosion through DEMs of Differences (DOD) (thus the literature suggested, including also (Lane et al. 2003) in addition to (Wheaton et al. 2009; Wheaton et al. 2010).

*We have included the four references on DEM differencing suggested by Referee 1, Lorenzo Marchi. There are now 5 references for this method.*

I also agree with the fact that the method part should be clarified more.

*We have clarified the methods more, see the various responses to the referees' comments on methods and the revised manuscript.*

As raised also during the review, i have one more question about the channel width size, I understand channel width is calculated from a regional regression based on drainage area (please cite such equation), however what is the goodness of the fitting of such equation to actual field-surveyed channel size in your study area? could you provide this information? This is important, since the proposed channel buffer width is dependent on the bankfull channel width.

*We only use the regional regression for bankfull width only as one that scales regionally with drainage area to establish a buffer surrounding the incised channels, not to define the channel edges. We do not have any field surveys of channel width. Alternatives of hand digitizing the incised channel forms or developing code to automate the procedure were not feasible considering the massive effort needed. We did try various factors of bankfull widths as buffers (2x, 4x, 6x) and checked them in the field and overlays of shaded relief and air photos to see which buffer fit best.*

*We have added this discussion to the text here including a new figure (Fig. 3) that gives an example of the buffer fit: "To use a buffer width that scales regionally with drainage area, we used a GIS buffer width of 6 times the total bankfull channel width that captures the steep eroding banks of the incised channel form and areas immediately adjacent to the channel (Fig. 3). To estimate bankfull channel width, we used a San Francisco Bay Area regional regression relationship based on drainage area (Dunne and Leopold 1978):*

$$bankfull\ channel\ width\ (m)\ =\ 3.3494\ (drainage\ area\ [km^2])^{0.3737} \qquad (1)$$

*In comparison to field measurements downstream from Arroyo Mocho, Bigelow et al. 2008 found the regional regression slightly over estimated actual bankfull width. The channel width estimate used here is only to create a buffer capturing the area around incised channels, not to*

*define channel edges. We checked the fit of several buffer sizes (2, 4, and 6 times total bankfull width) on the shaded relief DEM (Fig. 3), air photos, and in the field, and found the 6 times total bankfull width buffer was best suited to maximize the inclusion of both eroding banks of incised channels and hillslope areas that contribute sediment annually (e.g. toes of earthflows), but exclude areas that are not directly connected to channels. A small area of valley floor or hillslope of generally low erosion potential beyond the channel edge is included in this buffer. This buffering to capture the incised channel network still provides the most reasonable solution, considering the massive effort needed for alternatives to either digitize incised channels over a 573 km$^2$ watershed or develop code to automate the procedure....*"

The potential decoupling of the sediment respect to the network is also an interesting question raised during the review. I suggest the authors to consider for example (Cavalli et al. 2013, further investigated in Trevisani and Cavalli 2016) as another example to account for sediment connectivity.

*We have added the Cavalli et al. 2013 reference to the text as another example on how to estimate the proportion of sediment delivered to channels when analyzing sediment sources further from the channel.*

As one of the reviewers underlined, also the description of the channel network delineation needs clarification.

*We have added the major steps of the channel extraction method.*

You provided a series of replies which give a first overview of the steps you are going to take for the review. If you are willing to pursue these revisions, I will be pleased to reconsider your submission, with the help of the same reviewers who examined the present work. In submitting your revised version, please provide a detailed list of the changes made to the text, and a detailed list of your responses to each reviewer's comment. Please note that this editorial decision does not guarantee that your paper will be accepted for final publication in ESurf. A decision will be made only when the revised version will be available, and will be evaluated. Best regards Giulia Sofia

*Thank you for help in revising the manuscript, we look forward to hearing from you.*

Cavalli M, Trevisani S, Comiti F, Marchi L (2013) Geomorphometric assessment of spa- tial sediment connectivity in small Alpine catchments. Geomorphology 188:31–41. doi: 10.1016/j.geomorph.2012.05.007 Lane SN, Westaway RM, Hicks DM (2003) Estima- tion of erosion and deposition volumes in a large, gravel-bed, braided river using synop- tic remote sensing. Earth Surf Process Landforms 28:249–271. doi: 10.1002/esp.483 Trevisani S, Cavalli M (2016) Topography-based flow-directional roughness: potential and challenges. Earth Surf Dyn 4:343–358. doi: 10.5194/esurf-4-343-2016 Wheaton JM, Brasington J, Darby SE, Sear DA (2009) Accounting for uncertainty in DEMs from repeat topographic surveys: improved sediment budgets. Earth Surf Process Land- forms n/a–n/a. doi: 10.1002/esp.1886 Wheaton JM, Brasington J, Darby SE, Sear DA (2010) Accounting for uncertainty in DEMs from repeat

topographic surveys: Improved sediment budgets. Earth Surf Process Landforms 35:136–156. doi: 10.1002/esp.1886

---

## Author Response (AR2)

**Authors' response to Associate Editor and referee comments (dated 15 June 2016) on "Delineating incised stream sediment sources within a San Francisco Bay tributary basin"**

Responses are *italicized*

Comments to the Author:
Dear authors, I have now examined the second round of reviews of your paper. Both reviewers agree that the paper was greatly improved, and you satisfactorily answered to all the raised points. However, some minor issues still remain. The requested minor revision of the manuscript will be reviewed at the editorial level before acceptance.

Thank you for submitting to the Special Issue, and I look forward to receiving the revised manuscript.
Kind Regards.

Giulia Sofia

*Thank you Giulia and thanks also to the referees for all your help in revising this manuscript. We have made the revisions noted below that are highlighted in red text in the revised manuscript. We also made a few minor editorial changes to the manuscript, also in red text. We added 5 lines of text (in red) on page 11 lines 23-27 that describe a few more approaches for promoting sediment storage within incised channels and initial approaches used in Arroyo Mocho: "Where appropriate, sediment storage from incised streams could also be promoted with check dams (Geyik 1986) and stone weirs (Shields et al. 1995), or more natural analogs using large wood (Shields et al. 2003), beaver dams (Pollack et al. 2014), and Native American techniques (Norton et al. 1995). In the Arroyo Mocho watershed, initial strategies to reduce erosion from incised channels include construction of a step pool channel (Chin et al. 2009), sediment retention ponds, and earthen check dams that also serve as cattle ponds (Bigelow et al. 2012a)."*

Non-public comments to the Author:
Reviewer #2 rises a point about the methods description, where you write that "the contour length crossed by flow out of a DEM cell, which is used to calculate local contributing area". the reviewer thinks that this is not 100% correct - the contour length is needed together with the flow accumulation dataset in order to compute the specific contributing area. Could you clarify?

*The referee is correct, we have revised this sentence to read: "The attributes calculated are surface gradient, the number of adjacent inflowing DEM cells, and the contour length crossed by flow out of a DEM cell, where the latter two attributes are used to calculate specific contributing area (see details in next section)."*

in addition there is a typo at line 26 of page 3: "Define Sediment Source Area" instead of "Definine Sediment Source Area".

*We have corrected this typo in the revised manuscript.*